# Nickel-molybdenum nitride nanoplate electrocatalysts for concurrent electrolytic hydrogen and formate productions

Yan Li[1], Xinfa Wei[1], Lisong Chen [1]*, Jianlin Shi [1,2]* & Mingyuan He[1]

Hydrogen production by electrocatalytic water splitting is an efficient and economical technology, however, is severely impeded by the kinetic-sluggish and low value-added anodic oxygen evolution reaction. Here we report the nickel-molybdenum-nitride nanoplates loaded on carbon fiber cloth (Ni-Mo-N/CFC), for the concurrent electrolytic productions of high-purity hydrogen at the cathode and value-added formate at the anode in low-cost alkaline glycerol solutions. Especially, when equipped with Ni-Mo-N/CFC at both anode and cathode, the established electrolyzer requires as low as 1.36 V of cell voltage to achieve 10 mA cm$^{-2}$, which is 260 mV lower than that in alkaline aqueous solution. Moreover, high Faraday efficiencies of 99.7% for H$_2$ evolution and 95.0% for formate production have been obtained. Based on the excellent electrochemical performances of Ni-Mo-N/CFC, electrolytic H$_2$ and formate productions from the alkaline glycerol solutions are an energy-efficient and promising technology for the renewable and clean energy supply in the future.

[1] Shanghai Key Laboratory of Green Chemistry and Chemical Processes, School of Chemistry and Molecular Engineering, East China Normal University, Shanghai 200062, P. R. China. [2] State Key Laboratory of High Performance Ceramics and Superfine Microstructures, Shanghai Institute of Ceramics, Chinese Academy of Sciences, Shanghai 200050, P. R. China. *email: lschen@chem.ecnu.edu.cn; jlshi@mail.sic.ac.cn

The development of clean and renewable energy resources, such as solar, wind, and hydrogen energy, has attracted much attention because of the increasingly serious environment contamination and energy crises[1–3]. Molecular hydrogen is an attractive and carbon-neutral clean energy carrier for the future[4,5]. However, currently the most widely used process for hydrogen production is steam methane reforming, which is unsustainable due to the highly energy consumption by using the fossil fuels and the simultaneous $CO_{(2)}$ emissions as a byproduct, severely unfavorable for its applications[6]. Electrocatalytic water splitting is a simple and efficient technology, which can generate hydrogen with high-purity up to 100%, and has been considered as an attractive alternative in the future hydrogen economy[7,8]. However, both the cathodic hydrogen evolution reaction (HER) and anodic oxygen evolution reaction (OER) require large overpotentials, resulting in low energy conversion efficiency[9]. Although the theoretical minimum voltage is only 1.23 V for water splitting, commercial electrolyzers can efficiently work at voltages usually higher than 1.80 V[9,10]. Moreover, during water splitting, $O_2$ produced at the anode, which is relatively less valuable, will be unavoidably mixed with the $H_2$ produced at the cathode, implying a potential but highly dangerous explosion hazard, even high-cost membranes between the anode and the cathode are used[11,12]. In addition, the reactive oxygen species formed in the presence of $H_2$, $O_2$, and catalysts will most probably degrade the membrane and reduce the lifetime of the electrolyzer[13].

In view of the above considerations, a strategy that can lower the cell voltage input for electrolytic hydrogen production has been developed recently by chemical-assisted water splitting[14], in which more easily oxidized species (such as urea[15–17], ammonia[18], hydrazine[19], and methanol[12]) are added as sacrificial agents for electrochemical oxidation to replace the OER process. Although this strategy is able to lower the potential for water splitting, these high-cost sacrificial agents used would inevitably increases the total cost of hydrogen production. Glycerol, a kind of small organic molecules that can be more easily electro-oxidized than water, can be considered as a desirable alternative agent for OER in clean $H_2$ production[20,21]. Glycerol is a low-value added by-product during the production of biodiesel (about 10 wt % of the total product), and its selective oxidation has always been a hot research topic[22]. It has been demonstrated that various types of commodity chemicals can be produced by electrochemical conversion of glycerol, such as dihydroxy-acetone, glyceric acid, glycollic acid, and formic acid[23–26]. Among the products of glycerol oxidation, formic acid (HCOOH) or formate salts ($HCOO^-$) is a kind of important industrial intermediate or product because of their very high-added value in chemical industry. It has attracted great interest in the field of fuel cell applications and can be used for hydrogen storage due to its relatively high hydrogen capacity[27,28]. Unfortunately, traditional industrial methods for producing formic acid, methyl formate hydrolysis, are usually a complicated multistep process that consumes a large amount of energy, and the raw materials (e.g., toxic CO) used in this process usually from the gasification of coal and natural gas at elevated temperatures[29], which leads to the high cost of formic acid production. Considering the low value of glycerol, electrochemical glycerol oxidation has great potential for formic acid production. In addition, the complete oxidation of one molecule of glycerol to formic acid requires a theoretical oxidation potential of 0.69 V (vs. the standard hydrogen electrode; Supplementary Note 1), which is far lower than that of 1.23 V for OER under the standard conditions, the latter is the rate-determining step of the overall water electrocatalytic splitting process. Therefore, if glycerol can be electro-oxidized to formate in alkaline environment, the substitution for OER by glycerol oxidation will be a highly promising strategy which can not only reduce the cell voltage input for the production of $H_2$ at the cathode, but also produce more valuable formate at the anode at the consumption of low value glycerol. Unfortunately, up to now, noble metals and their oxides are still the most popular electrocatalysts for the HER and glycerol oxidation. Suffering from the high cost, scarcity and poor stability of precious metal catalysts, the large scale production is still far away from industrialization[30,31], which makes the development of earth-abundant, low-cost, high performance, and stable non-noble-metal bifunctional electrocatalyst extremely significant and attractive for both efficient electro-catalytic glycerol oxidation and HER. However, searching for such electro-catalysts remains greatly challenging presently, and no reports can be found on such electro-catalysts for $H_2$ generation assisted by glycerol.

Transition metal-based nitrides, which usually have relatively low electrical resistance and high mechanical stability, are considered as promising electrocatalysts and have shown enhanced activities in a variety of reactions such as OER[32], HER[33], and ORR[34]. Here, we report an efficient non-noble metal electrocatalyst, nickel–molybdenum–nitride nanoplates on carbon fiber cloth (Ni–Mo–N/CFC) at both cathode and anode, for the concurrent high purity hydrogen and value-added formate productions from glycerol aqueous solution under the alkaline condition. In this work, the oxidation of glycerol to much more valuable formate instead of OER at the anode catalyzed by Ni–Mo–N/CFC, and simultaneously donates electrons to reduce water and generate $H_2$ under the catalysis also by Ni–Mo–N/CFC at the cathode, in which both reactions are highly efficient without using membrane. A potential of as low as 1.30 V is required for the glycerol oxidation at anode to reach a geometrical catalytic current density of 10 mA cm$^{-2}$. When the electrocatalyst is integrated into a two-electrode alkaline electrolyzer, the current density of 10 mA cm$^{-2}$ can be obtained at a rather low cell voltage of 1.36 V, which is 260 mV less than that required for electrolytic pure water splitting. This strategy utilizing a non-noble metal electrocatalyst to produce high-value products at the cathode and anode is believed to be of great significance in the development of renewable-energy technology.

## Results and Discussion

**Materials characterization**. The Ni–Mo–N/CFC nanoplates were synthesized facilely by the growth of NiMo-precursor on conductive carbon fiber cloth (NiMo-Pre/CFC) through a hydrothermal reaction, followed by the precursor annealing under an ammonia gas to obtain the Ni–Mo–N/CFC nanoplates (see the Methods for detailed synthetic approaches). The morphology of the obtained NiMo-Pre/CFC was observed by scanning electron microscopy (SEM). As shown in Supplementary Fig. 1, the carbon fiber cloth is compactly covered by smooth nanoplates, which are about 677 nm in size and 73.0 nm in thickness (Supplementary Fig. 1a, b). The X-ray diffraction (XRD) pattern (Supplementary Fig. 2) shows that the NiMo-Pre/CFC contains $MoO_3 \cdot 0.5H_2O$ (no. 49–0652), $MoO_3 \cdot H_2O$ (no. 28-0666), $MoO_2$ (no. 32-0671), and NiO (no. 44-1159). After nitridation treatment at 500 °C, the Ni–Mo–N/CFC nanoplates, which were derived from the NiMo-Pre/CFC, are composed of mixed phases of major Ni (no. 04-0850) and minor $Ni_{0.2}Mo_{0.8}N$ (no. 29-0931), as confirmed by the XRD pattern (Fig. 1a). SEM images show that the Ni–Mo–N/CFC nanoplates are approximately 682 nm in diameter and 57.9 nm in thickness, and the surface become rougher at elevated annealing temperature from 400 to 600 °C (Fig. 1b, c and Supplementary Fig. 3). In more detail, the transmission electron microscopy (TEM) image (Fig. 1d) reveals that the nanoplate structure is composed of nanoparticles ~15 nm in average diameter. As

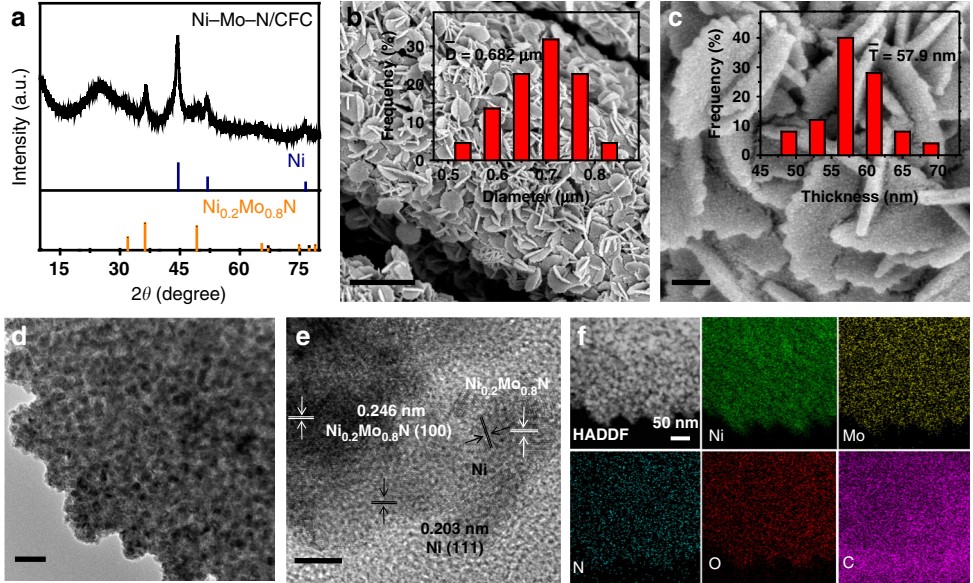

**Fig. 1** Structural and morphological characterizations of Ni–Mo–N/CFC. **a** XRD pattern, **b**, **c** SEM, **d** TEM, **e** HRTEM images, and **f** STEM-EDX mapping of Ni–Mo–N/CFC. The inset images in **b** and **c** are the diameter (left) and thickness (right) distributions of Ni–Mo–N/CFC nanoplates. Scale bars, **b** 2 μm; **c** 200 nm; **d** 50 nm; **e** 5 nm, **f** 50 nm.

shown in the high-resolution TEM (HRTEM) image, the visible lattice d-spaces of Ni–Mo–N/CFC are around 0.203 and 0.246 nm, which can be indexed to the (111) lattice planes of Ni and (100) lattice planes of $Ni_{0.2}Mo_{0.8}N$, respectively (Fig. 1e). Furthermore, scanning transmission electron microscopic high-angle annular dark field imaging and corresponding elemental mapping (Fig. 1f) show the uniform distributions of Ni, Mo, N, O, and C elements (the detected C element is attributed to CFC and carbon support films used in TEM measurements), which is in accordance with the results of the SEM elemental mapping images (Supplementary Fig. 4), indicating the successful transformation of NiMo-Pre/CFC to Ni–Mo–N/CFC nanoplate product.

To further investigate the surface chemical composition and valence states, the X-ray photoelectron spectroscopy (XPS) was adopted. The mass ratio of nickel to molybdenum from XPS is about 1:1.90, which is in line with the result of inductively coupled plasma optical emission spectrometric (ICP-OES) measurements. Supplementary Fig. 5a shows the survey spectrum of Ni, Mo, N, O, and C elements of Ni–Mo–N/CFC. The high-resolution $Ni2p_{3/2}$ spectrum (Supplementary Fig. 5b) of Ni–Mo–N/CFC reveals the presences of metallic Ni and $Ni^{2+}$ at 852.9 and 855.8 eV, respectively[35,36]. The Mo 3d spectrum (Supplementary Fig. 5c) can be deconvolved into six subpeaks, including $Mo^{3+}$ (229.42 and 232.66 eV), $Mo^{4+}$ (230.5 and 233.64 eV), and $Mo^{6+}$ (232.12 and 235.45 eV) species[37–39]. $Mo^{3+}$ is a component in $Ni_{0.2}Mo_{0.8}N$, and has been reported to be active for HER[40]. $Mo^{4+}$ and $Mo^{6+}$ originate from the surface oxidization upon air exposure[41]. In the N 1s-Mo 3p spectrum (partial overlap of N 1s and Mo 3p) (Supplementary Fig. 5d), the peaks at 395.3, 397.47, and 399.41 eV observed in the Ni–Mo–N/CFC can be assigned to Mo $3p_{3/2}$, metal–N bonds and N–H groups, respectively[42,43]. The O 1s XPS spectrum can be deconvoluted into three major peaks, as depicted in Supplementary Fig. 5e. The peak at 530.2 eV could be attributed to metal–oxygen bonds in the metal oxide[44]. The peak at 531.2 eV can be assigned to oxygen in –OH groups, suggesting that the surface of the Ni–Mo–N/CFC has been hydroxylated, and the peak located at 532.6 eV is likely associated with oxygen from carbonyl groups in the CFC[45,46]. As for the C 1s spectrum, three contributions appearing at 284.6, 285.79, and 287.8 eV can be observed, assignable to C–C/C = C,

C–OH, and C = O from CFC[47–49], respectively. Moreover, the pure CFC was also characterized by XPS (Supplementary Fig. 6) and the chemical status of carbon is almost identical with those of Ni–Mo–N/CFC electrode in the C1s spectrum, indicating negligible effect of the synthetic processes on the CFC.

**Electrocatalytic performances for glycerol oxidation**. To investigate the electrocatalytic performance of as-prepared Ni–Mo–N/CFC for glycerol oxidation at the anode, a series of electrochemical measurements were performed in a three-electrode setup. Figure 2a presents the linear sweep voltammetry (LSV) curves of Ni–Mo–N/CFC in 1.0 M KOH with and without 0.1 M glycerol. In the absence of glycerol, the electrode shows a moderate OER activity, which reaches the anodic current density of 10 mA cm$^{-2}$ at the potential of 1.57 V versus reversible hydrogen electrode (vs. RHE). The peak centered at about 1.4 V vs. RHE for the OER can be ascribed to the oxidation peak of $Ni^{2+}/Ni^{3+}$[50]. After introducing 0.1 M glycerol, the current density increased markedly, and the anodic potential strikingly decreased to 1.30 V vs. RHE at 10 mA cm$^{-2}$ (3 successive LSVs in the Supplementary Fig. 7 and cyclic voltammogram curves in the Supplementary Fig. 8). To give a more detailed comparison with OER, Fig. 3b displays that the anodic potentials in glycerol solution are reduced by at least 250 mV at the current densities of 10, 20, 50, 100, and 150 mA cm$^{-2}$, which are lower than those of most reported systems for electro-chemical-assisted water splitting, except for several strong reducing agents such as hydrazine for chemical-assisted water splitting (Supplementary Table 4). Tafel slope was derived from the LSV data to evaluate the reaction kinetics as shown in Fig. 2c. The Ni–Mo–N/CFC shows a rather low Tafel slope of 87 mV dec$^{-1}$ for the glycerol oxidation, which is substantially lower than that of OER (161 mV dec$^{-1}$), indicating a much faster catalytic kinetics at the anode. The effect of glycerol concentration in electrolyte on the glycerol oxidation performance at the anode has also been investigated, and the concentration of glycerol has been optimized as given in Supplementary Fig. 9. The highest glycerol oxidation activity was achieved in 1 M KOH solution containing 0.1 M glycerol. Noticeably, Ni–Mo–N/CFC shows significantly high performance than those of the CFC, NiMo-Pre/CFC, Ni/CFC (Ni nanoparticles

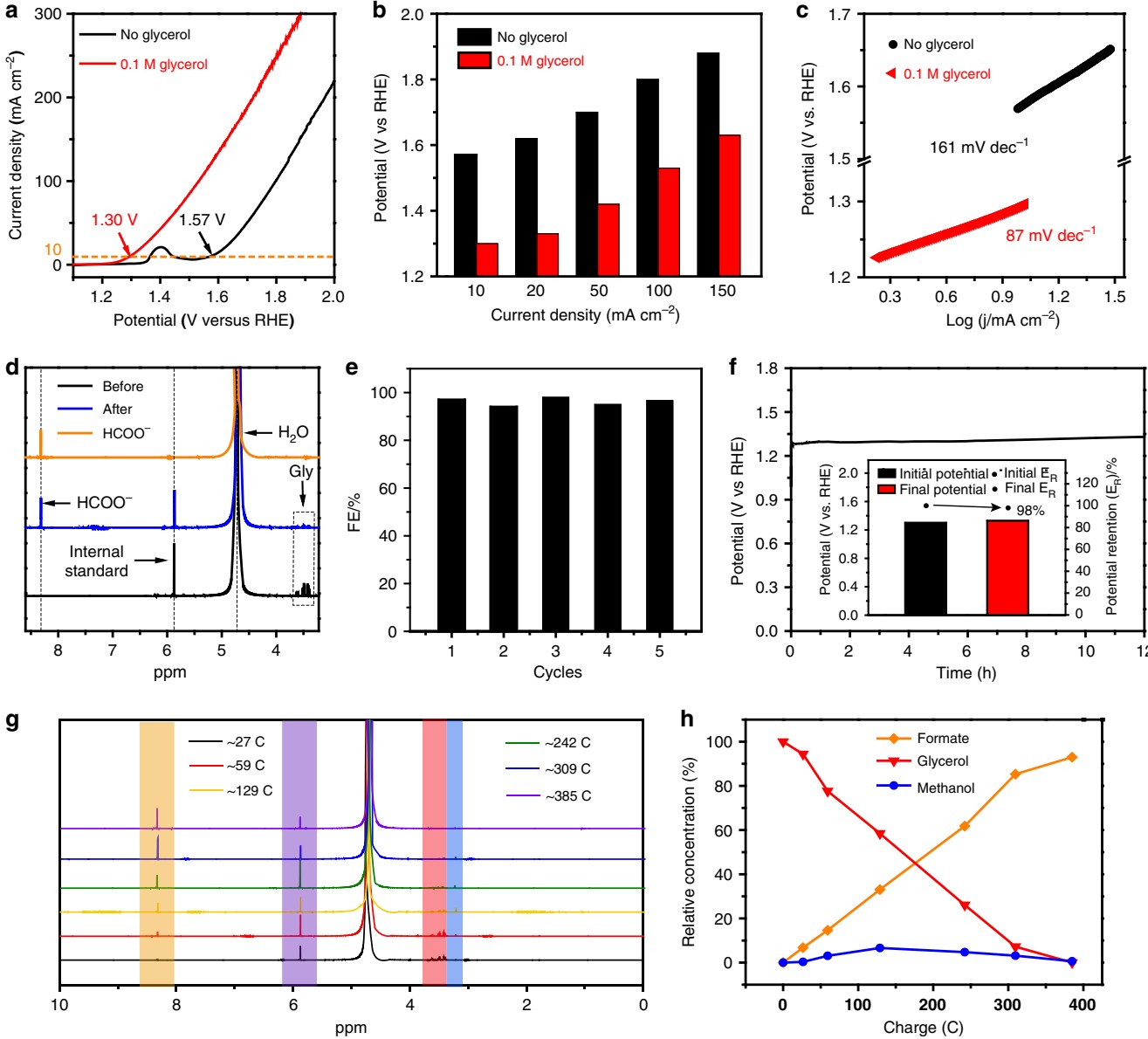

**Fig. 2** Electrocatalytic performances of Ni–Mo–N/CFC in glycerol oxidation at the anode. **a** LSV curves of Ni–Mo–N/CFC anode in 1.0 M KOH with and without 0.1 M glycerol (Gly) addition. Scan rate, 2 mV s$^{-1}$. **b** Comparisons of the anodic potentials to achieve varied current densities for Ni–Mo–N/CFC in 1 M KOH with and without 0.1 M glycerol addition. **c** Tafel plots for the anodic glycerol oxidation or OER derived from (**a**). **d** $^1$H NMR spectra of products before and after 12 h anodic glycerol oxidation on Ni–Mo–N/CFC electrode by using Maleic acid as the internal standard, and that of standard formate, demonstrating the conversion of most glycerol to formate. **e** Faradaic efficiencies (FEs) of Ni–Mo–N/CFC for formate production for five successive electrolysis cycles. **f** Twelve hour stability test of Ni–Mo–N/CFC for the anodic glycerol oxidation at a current density of 10 mA cm$^{-2}$. The inset image in **f** shows the changes of the potentials during chronopotentiometric glycerol oxidation tests (bar graphs). **g** $^1$H NMR measurements of glycerol oxidation of glycerol (red) to formate (orange) and methanol (blue). Maleic acid was added as an internal standard (purple). **h** Glycerol consumption into formate and methanol.

loaded on carbon fiber cloth, Supplementary Figs. 10 and 11) and IrO$_2$/CFC reference for glycerol oxidation (Supplementary Fig. 12a). Besides, Ni–Mo–N/CFC shows the lowest Tafel plots (87 mV dec$^{-1}$) compared to NiMo-Pre/CFC (92 mV dec$^{-1}$), Ni/CFC (121 mV dec$^{-1}$), CFC (166 mV dec$^{-1}$), and IrO$_2$/CFC (214 mV dec$^{-1}$), as shown in Supplementary Fig. 12b, which indicates the fastest reaction kinetics. According to the electrochemical impendence spectra (EIS) and the corresponding fitting results, Ni–Mo–N/CFC electrode possesses a much smaller charge transfer resistance than those of NiMo-Pre/CFC and CFC, further suggesting an excellent electrical contact, extremely low electric impedance and fast charge transfer rate (Supplementary Fig. 13).

It should be mentioned that with the increase of calcination temperature, the activity of glycerol oxidation shows no significant change (Supplementary Fig. 14a). Moreover, as can be seen in Supplementary Fig. 14, Ni–Mo–N/CFC shows a much higher electrochemical double-layer capacitance ($C_{dl}$) value (0.302 F cm$^{-2}$) than Ni–Mo–N/CFC-400 (0.215 F cm$^{-2}$), Ni–Mo–N/CFC-600 (0.037 F cm$^{-2}$), and NiMo-Pre/CFC (0.218 F cm$^{-2}$), indicating the largest electrochemically active surface areas (ECSA) and most exposed active sites.

Subsequently, a long-term chronoamperometry test using Ni–Mo–N/CFC as the electrocatalyst was carried out at 1.35 V vs. RHE to qualitatively and quantitatively determine the

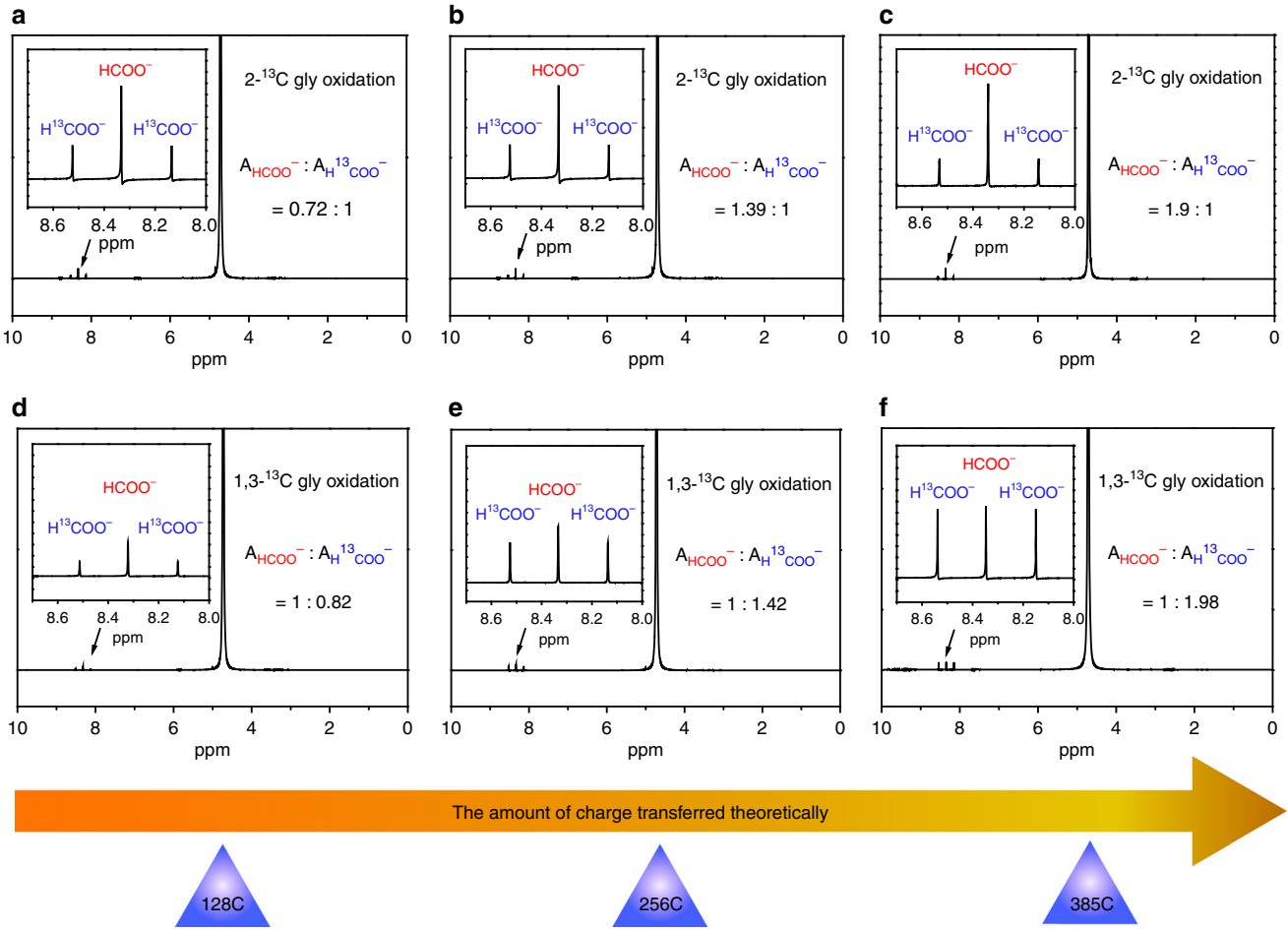

**Fig. 3** $^1$H NMR spectra for the electro-oxidation of 2-$^{13}$C-glycerol and 1, 3-$^{13}$C-glycerol. **a–c** 0.1 M, 2-$^{13}$C-glycerol conversions theoretically into ~0.1 M (**a**), ~0.2 M (**b**) and ~0.3 M (**c**) formate at 128, 256, and 385 C charges transferred, respectively; **d–f** 0.1 M 1, 3-$^{13}$C-glycerol conversions theoretically into ~0.1 M (**d**), ~0.2 M (**e**), and ~0.3 M (**f**) formate at 128, 256, and 385 C charges transferred, respectively. $A_{HCOO^-}$ and $A_{H^{13}COO^-}$ are the integral area of HCOO$^-$ and H$^{13}$COO$^-$ in $^1$H NMR spectra, respectively.

obtained glycerol oxidation products at the anode by the $^1$H, $^{13}$C NMR spectroscopy and ion chromatograph analyses. As shown in Supplementary Fig. 15, the C–C bond cannot be completely broken down at relatively low potentials and OER will only take place at elevated potential, so, a potential of ~1.35 V was selected as the optimum potential without any other special comments, resulting in both high selectivity (~92.48%) and Faradaic efficiency (~97 %). The $^1$H NMR spectra of the product reveals the presence of formate besides H$_2$O, the internal standard, glycerol, while no other glycerol oxidation products can be found (Fig. 2d). Both formate and carbonate can be detected after 12 h glycerol oxidation by $^{13}$C NMR spectroscopy and ion chromatograph, but the Faradic efficiency for the detected carbonate is extremely low at ~2.4% (the high solubility of CO$_2$ from air in alkaline medium may lead to some errors although efforts have been made to deduct the amount of carbonate in the electrolyte that is not derived from the electrochemical reaction), which is a very small value, suggesting a very limited amount of CO$_3^{2-}$ formation from glycerol (Supplementary Figs. 16 and 17). The ratio of transferred electrons to the molar amount of formate is 2.72:1–2.84:1, which is close to 8:3, implying that three moles of formate can be produced from every mole of glycerol consumed theoretically. The Faradaic efficiencies (FEs) of 94%~98% for formate production have been maintained by five successive cycles of Chronoamperometry (see details in Methods) (Fig. 2e). The above results indicate that formate is the main oxidation

product of glycerol, demonstrating excellent selectivity of the glycerol electro-oxidation to formate electro-catalyzed by Ni–Mo–N/CFC, which cannot be found using other reported catalysts. Moreover, the electrochemical stability of Ni–Mo–N/CFC electrode for glycerol oxidation was also evaluated by recording $U$–$t$ curves (Fig. 2f). No significant potential increase can be observed during the 12 h stability test, elucidating the robust durability of the anode.

**Mechanistic insight**. To further explore the reaction mechanisms, a long-time electrocatalysis experiment has been carried out to monitor the concentration variations of glycerol and product (formate) during the whole glycerol electrolysis period at 1.35 V vs. RHE. The concentration decrease of glycerol and the concentration increase of formate over time can be clearly seen in the Fig. 2g, indicating the efficient conversion of glycerol to formate. After 385 C electric charge was transferred through the electrocatalysis cell, the relative concentrations of formate and glycerol increased to the maximum and decreased to 0, respectively, which suggests the complete conversion of glycerol, leading to a yield of 93% for formate production (Fig. 2h). Surprisingly, methanol was undoubtedly detected and the ratio of methanol to formate increased to around 1:5 at ~129 C charges transferred, and then began to decrease until it cannot be detected. To explore the source of methanol, a three-electrode configuration using

**Fig. 4** Proposed mechanistic scheme of glycerol electro-catalytic oxidation to formate on Ni–Mo–N/CFC in alkaline medium (blue arrows is proposed to be the dominant pathways).

Ni–Mo–N/CFC catalyst was set up by adding 0.1 M formate dissolved in 1 M KOH at the cathode. No other products were detected except hydrogen at the cathode side (Supplementary Fig. 18), confirming that methanol does not come from the formate reduction of cathodic reaction. Fortunately, formaldehyde was detected via phloroglucinol method, a highly sensitive method for detecting formaldehyde with a detection limit of 0.1 ppm (Supplementary Fig. 19). Therefore, we infer that the methanol was from the Cannizzaro reaction (an aldehyde without an α-hydrogen atom undergoes an intermolecular redox reaction under the action of a strong base to form a carboxylic acid and an alcohol) of formaldehyde in alkaline solution[51,52]. Besides, we also found that the carbon atoms of methanol came from carbon atoms at positions 1, 3 of glycerol according to the results of isotope tracer described below.

To better understand the mechanism of glycerol electro-oxidation to formate, experiments using 0.1 M 1, 3-[13]C-labeled glycerol and 0.1 M 2-[13]C-labeled glycerol in 1 M KOH as electrolyte and Ni–Mo–N/CFC as electrocatalyst have been further performed at 1.35 V vs. RHE (Fig. 3). From the [1]H NMR analysis it can be found that the ratios of unlabeled formate to [13]C-labeled formate obtained by 2-[13]C-labeled glycerol oxidation are 0.72:1, 1.39:1, 1.9:1, and the corresponding ratios obtained by 1,3-[13]C-labeled glycerol oxidation are 1:0.82, 1:1.42, 1:1.98, respectively, during the progress of the glycerol oxidation reaction to varied stages, as determined by [1]H NMR analysis. Therefore, it is clear that the formed product, formate, comes from both the secondary and primary carbons of glycerol.

Moreover, the methanol can be detected in 1, 3-[13]C-labeled glycerol, while it cannot be detected in 2-[13]C-labeled glycerol by the [13]C NMR spectroscopy in monitoring the source of methanol (Supplementary Fig. 20), which means that the formation of formate is a consequence of successively stepped reaction after a long-time electrolysis with methanol being one of the major intermediates, as described and supposed in more detail in the following.

According to the above results and previous reports[24,53–59], a possible reaction path of glycerol electro-oxidation at the anode to formate is proposed as shown in Fig. 4. Firstly, the formation of formate begins with glycerol oxidation to glyceraldehyde, which is then oxidized to formate and glycolaldehyde with the breakage of the C–C bonds. Next, the oxidative cleavage of the glycolaldehyde produces formate and formaldehyde, followed by the methanol and formate formations by the intermolecular redox reactions (Cannizzaro reaction) of formaldehyde in alkaline solution, and the methanol final oxidation to formate. In all,

almost all of the reactant glycerol and several intermediates are eventually oxidized to formate. In this overall pathway of glycerol to formate, a very small amount of carbonate (~2.4%) may come from the further oxidation of formate. Furthermore, chronopotentiometric (CP) curves for Ni–Mo–N/CFC of glycerol oxidation at 0.1 M and lower concentrations were recorded (Supplementary Fig. 21) and it is clear that the potential become stabilized during the decrease of glycerol concentration from 0.1 to 0.01 M, likely due to the oxidation balance between glycerol and the methanol. The potential increased significantly on the CP curve of glycerol oxidation at 0.005 M glycerol beyond about 6 h, indicating the taking-place of water oxidation.

In addition, weak peaks of glycolate can be detected in the [13]C-labeled glycerol oxidation products by the [13]C NMR spectroscopy due to the labeling of [13]C though they cannot be detected in the unlabeled glycerol oxidation products, indicating that the successive oxidation of glycerol to glycerate, and then to formate, a typical pathway of the glycerol electro-oxidation to formate, is a minor side reaction pathway in the present study.

**Electrocatalytic performances for HER.** The as-prepared electrode also demonstrates prominent performance for HER when used as cathode in the alkaline electrolyte. As depicted in Fig. 5a and Supplementary Fig. 22a, the HER overpotential of Ni–Mo–N/CFC is 40 mV to deliver a current density of 10 mA cm$^{-2}$, which is lower than those of CFC, Ni/CFC, and NiMo-Pre/CFC. Specifically, the Ni–Mo–N/CFC Tafel slope (70 mV dec$^{-1}$) is rather close to that of precious metal catalyst Pt/C/CFC (58 mV dec$^{-1}$) and much lower than those of NiMo-Pre/CFC (208 mV dec$^{-1}$) and Ni/CFC (191 mV dec$^{-1}$), suggesting the favorable HER reaction kinetics of the Ni–Mo–N/CFC (Supplementary Fig. 22b). Meanwhile, the EIS analyses show the smallest charge transfer resistance of Ni–Mo–N/CFC, which directly contribute to its high catalytic activity (Supplementary Fig. 23). In addition, the relationship between HER activity and annealing temperature was also studied, and the sample annealed at 500 °C is of the highest HER activity (Supplementary Fig. 24a). To gain more insight into the HER electrocatalytic activity, the ECSA values of different samples were obtained by extracting the corresponding Cdl, and it is clear that Ni–Mo–N/CFC shows a larger ECSA than those of Ni–Mo–N/CFC-400 and Ni–Mo–N/CFC-600, though a little lower than NiMo-Pre/CFC (Supplementary Fig. 24). In the consideration of the glycerol tolerance of HER, the HER activity of Ni–Mo–N/CFC was assessed at varied glycerol concentrations, which were found to be comparable to the activity without glycerol being added in the electrolyte (Fig. 5a and Supplementary

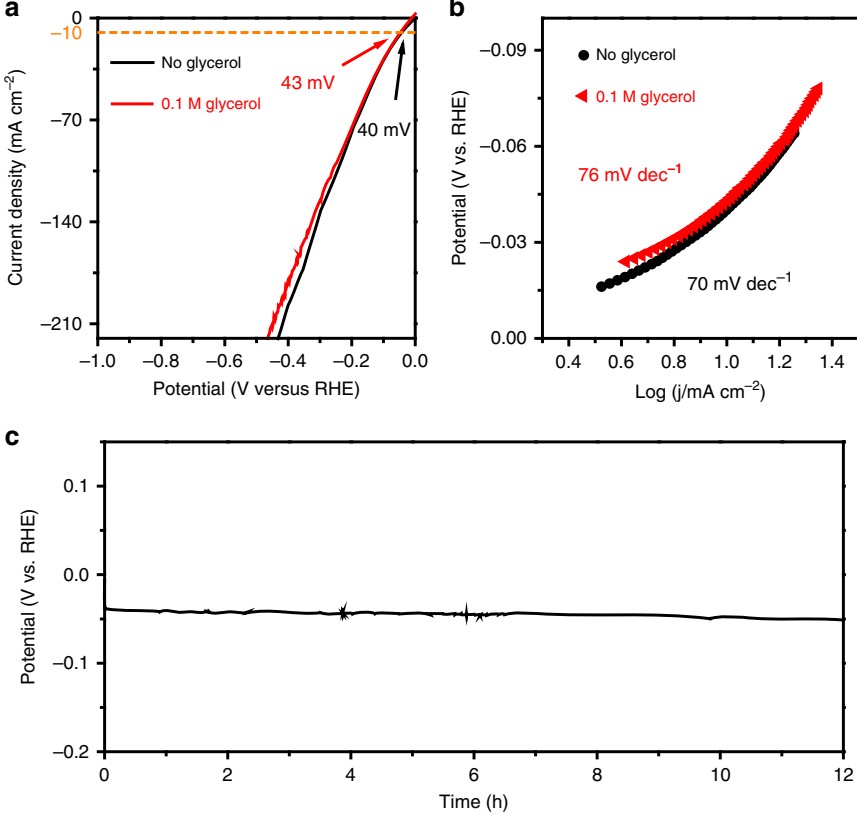

**Fig. 5** Electrocatalytic performances of Ni–Mo–N/CFC in HER at the cathod. **a** LSV curves for the HER of Ni–Mo–N/CFC in 1.0 M KOH with and without 0.1 M glycerol addition. Scan rate, 2 mV s$^{-1}$. **b** Tafel plots for the HER derived from (**a**). **c** Twelve hour stability test of Ni–Mo–N/CFC for cathodic HER at a current density of 10 mA cm$^{-2}$.

Fig. 25). As shown in Fig. 5b, the Tafel slope shows no significant changes after adding 0.1 M glycerol. Moreover, after 12 h of CP measurement at a constant current density of 10 mA cm$^{-2}$, the Ni–Mo–N/CFC maintains its excellent catalytic performance (Fig. 5c).

**Electrocatalysts characterizations after glycerol oxidation and HER.** In order to gain a better understanding on the catalytic and durability mechanisms of the HER and formate production, the Ni–Mo–N/CFC electrocatalysts and electrolytes after glycerol oxidation (post-GOR Ni–Mo–N/CFC) and HER (post-HER Ni–Mo–N/CFC) CP measurements were further characterized. As shown in Fig. 6a, the high-resolution Ni 2p$_{3/2}$ XPS spectra of the post-GOR Ni–Mo–N/CFC exhibit a clear intensity increase at 856.5 eV, suggesting the oxidations of most Ni (0) and/or Ni (II) to NiOOH (Supplementary Table 3), which has been reported to be the active species for glycerol electro-oxidation[60–62]. Accordingly, the peak fitting analysis of O1s shows a larger amount of –OH species at 531.6 eV (Supplementary Fig. 26a) than that of fresh Ni–Mo–N/CFC after glycerol oxidation CP, indicating the NiOOH formed by the substitution of oxygen atoms at the surface as well as the structural changes or the more formation of surface hydroxides[45]. Besides, Ni–Mo–N/CFC was found to lose more than 90% of molybdenum after glycerol oxidation at the anode, either in a short-time (20 cycles of CV scans + 3 LSV) or a long-time electrochemical test (20 cycles of CV scans + 3LSV + 12 h CP), as examined by ICP-OES analysis (Supplementary Table 1). The dissolved Mo could be detected in the electrolyte (Supplementary Table 2), and the Mo loss from the catalyst may lead to the formation of defects in the catalyst, thus resulting in more exposed active sites and enhanced electrocatalytic activity.

A similar situation that the loss of Mo could lead to defect generation has recently been described in a Chen's report[43]. In addition, no Mo species was found in the TEM images and XRD pattern of the post-GOR Ni–Mo–N/CFC, and extremely small amount metallic Ni can be found after glycerol oxidation at anode (Fig. 6d, e and Supplementary Fig.27 ). Furthermore, the residual Mo$^{3+}$ and Mo$^{4+}$ species were completely oxidized to Mo$^{6+}$ as shown in the high-resolution Mo 3d XPS spectra of post-GOR Ni–Mo–N/CFC (Supplementary Fig. 28a). In the N 1s-Mo 3p spectra (Supplementary Fig. 28b), the content of N elements in the form of metal–N species also decreased compared to the total N amount, which is in correspondence to the Mo loss mentioned above. The massive molybdenum was dissolved from the catalyst after an initial short-time electrochemical test, after that the catalyst with much decreased molybdenum amount maintains excellent stability in subsequent CP tests, which implies that Mo species is not active site for glycerol oxidation in this work. Different from the case of the anode, only a small amount of Mo was dissolved (Supplementary Tables 1 and 2) from the cathode Ni–Mo–N/CFC catalyst for HER and the valence states of Mo remain unchanged (Supplementary Fig. 28c). Besides, the content of N in the form of metal–N species decreased slightly (Supplementary Fig. 28d) and the Ni–Mo–N/CFC maintained its crystalline structure (Supplementary Fig. 27). The substantially different behaviors of the catalyst between anode and cathode is supposed to result from the fact that the Mo$^{6+}$ species can be easily dissolved in alkaline solution in the forms of molybdate and polymolybdate in the initial test period. Almost all Mo species can be oxidized to Mo$^{6+}$ in oxidizing conditions at anode, leading to a majority of Mo being leached into the electrolyte. In contrast, under the reduction conditions, only a minor amount of Mo

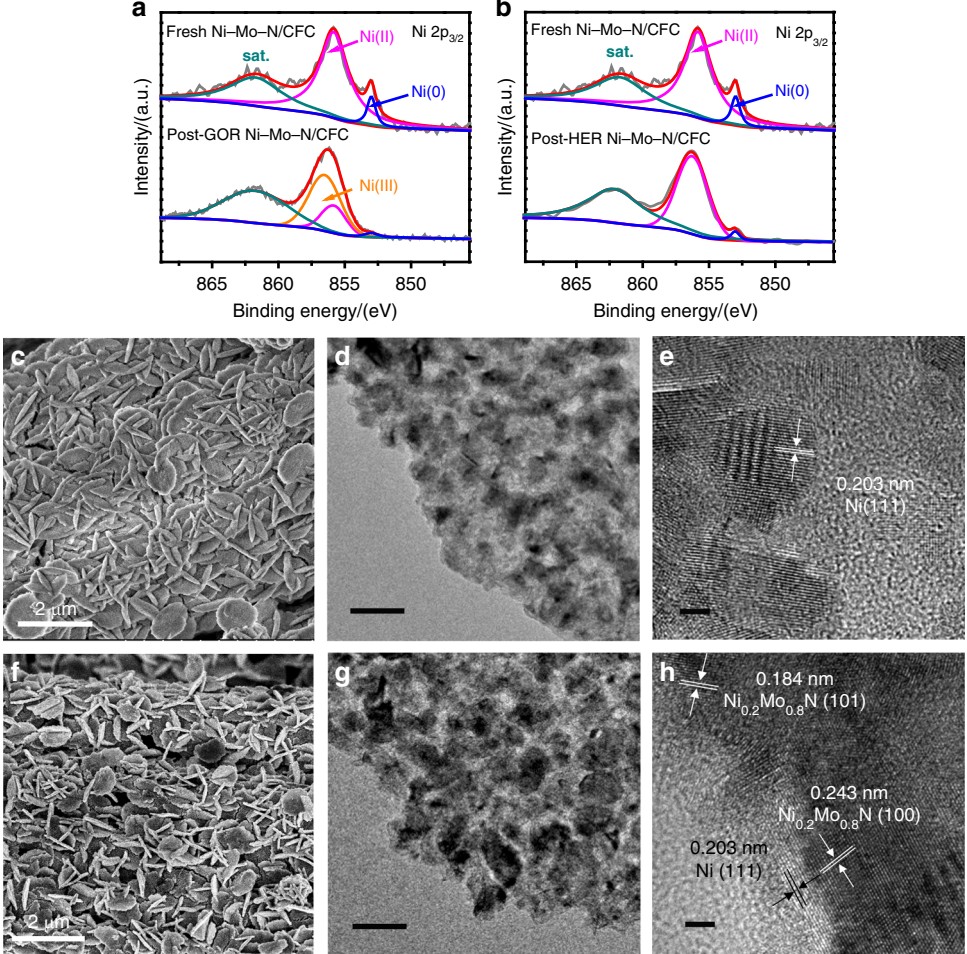

**Fig. 6** Morphology and composition characterizations of Ni–Mo–N/CFC after glycerol oxidation and HER. **a** High-resolution Ni $2p_{3/2}$ XPS spectra of Ni–Mo–N/CFC before and after glycerol oxidation at the anode. **b** High-resolution Ni $2p_{3/2}$ XPS spectra of Ni–Mo–N/CFC before and after HER at the cathode. **c** SEM, **d** TEM, and **e** HRTEM images of the anodic post-GOR Ni–Mo–N/CFC. **f** SEM, **g** TEM, and **h** HRTEM images of cathodic post-HER Ni–Mo–N/CFC. Scale bars, **c** 2 μm; **d** 50 nm; **e** 2 nm; **f** 2 μm; **g** 50 nm, **h** 2 nm.

species were converted to $Mo^{6+}$ species due to exposure to air, which leads to only a very limited amount of Mo loss at the cathode. In addition, most of Ni (0) on the surface of cathodic Ni–Mo–N/CFC has been converted to Ni (II) (Supplementary Table 3), probably due to the presence of water vapor as demonstrated by Sargent's group, or dissolved by oxygen in the electrolyte[63,64], which has been proved to be the active site for $H_2$ evolution from water decomposition (Fig. 6b)[65]. NH groups are present in the samples before and after the HER processes, which also facilities HER[66]. Moreover, the O 1s spectra after HER (Supplementary Fig. 26b) have also been obtained and three peaks can be observed corresponding to the metal–oxygen bonding, hydroxylation and carbonyl groups, respectively. In addition, the electrocatalytic processes both at the anode and the cathode have negligible effect on the surface chemical status of CFCs, suggesting the only conductive role of the CFC substrate (Supplementary Fig. 29). Apparently, the Ni–Mo–N/CFC maintained its original nanoplate morphology after both glycerol oxidation and HER stability tests (Fig. 6c, f–h), though molybdenum species has been largely dissolved, or slightly dissolved, at anode or cathode, respectively, contributing to the great durability of the catalyst.

**Electrochemical performances of Ni–Mo–N/CFC ‖ Ni–Mo–N/CFC**. Based on the excellent electrochemical performance of the

Ni–Mo–N/CFC for glycerol oxidation at the anode and HER at the cathode, a two-electrode electrolyzer with Ni–Mo–N/CFC as both anode and cathode catalysts (denoted as Ni–Mo–N/CFC ‖ Ni–Mo–N/CFC) and 1 M KOH solution containing 0.1 M glycerol as electrolyte for the concurrent electrochemical hydrogen and formate productions has been set up and operated at room temperature (Fig. 7a), which shows a potential of as low as 1.36 V required at the current density of 10 mA cm$^{-2}$ (Fig. 7b), exhibiting excellent HER/glycerol oxidation performance concurrently. Comparatively, a conventional overall water splitting electrolyzer in the same Ni–Mo–N/CFC ‖ Ni–Mo–N/CFC cell but without glycerol addition in the electrolyte was also tested under the same conditions, in which a much higher potential of 1.62 V was necessary to obtain the identical current density. Such a low cell voltage of the Ni–Mo–N/CFC for chemical-assisted water splitting is much lower than those of most reported systems (Supplementary Table 4). In addition, this kind of electrolyzer is of a long-term operational stability as depicted in Fig. 7c. As shown in Fig. 7d, the average FE of $HCOO^-$ formation is as high as 95.0% at 1.4 V (no noticeable competing from water oxidation), and accordingly, the amount of experimentally measured $H_2$ production is in good agreement with the theoretical value, from which the FE of HER has been calculated to be 99.7% (Fig. 7e). To further explore the rationality of the present experimental device of using no separating membrane, a cell of two chambers

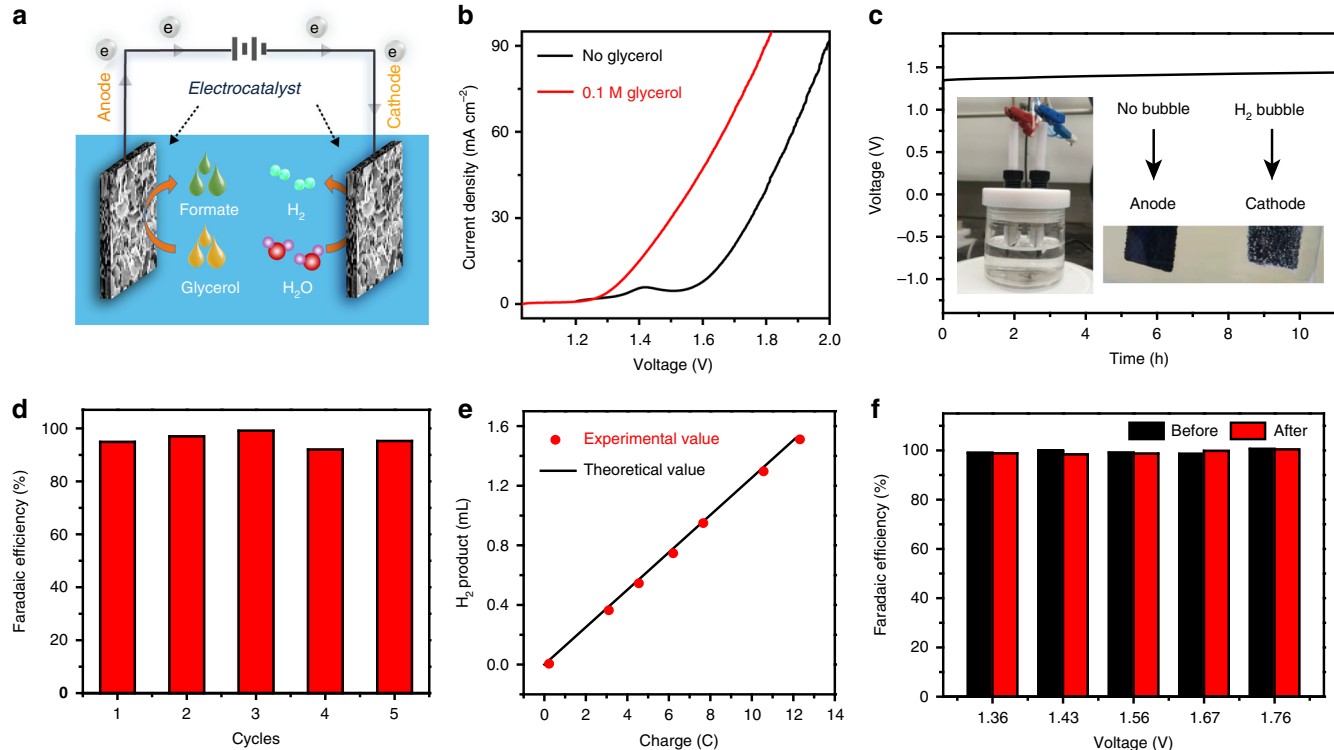

**Fig. 7** Electrochemical performances of Ni–Mo–N/CFC || Ni–Mo–N/CFC. **a** Schematic illustration for concurrent electrolytic hydrogen and formate productions from glycerol solution. **b** LSV curves for the Ni–Mo–N/CFC || Ni–Mo–N/CFC cell in 1.0 M KOH solution with and without 0.1 M glycerol addition. Scan rate, 2 mV s$^{-1}$. **c** Stability test of the Ni–Mo–N/CFC couple in the cell at a constant current density of 10 mA cm$^{-2}$ in 1.0 M KOH containing 0.1 M glycerol. **d** FEs of the Ni–Mo–N/CFC couple for formate production for five successive electrolysis cycles. **e** Comparison between theoretical calculation and experimental measurement of H$_2$ evolution, indicating an up to 100% FE for H$_2$ production on Ni–Mo–N/CFC catalysts. **f** FEs of the Ni–Mo–N/CFC couple for H$_2$ at different potentials from the beginning to the end of 12 h glycerol electrolysis.

separated by a Nafion membrane for the concurrent electro-chemical hydrogen and formate productions was set up by adding 0.1 M glycerol dissolved in 1 M KOH to the anode and 0.1 M formate dissolved in 1 M KOH to the cathode compartment. It is clear that the Faraday efficiencies for hydrogen at varied current densities are all consistently close to 100% (Supplementary Fig. 30a). The products in the anode are identical with that of the membrane-free system, and only formate can be detected in the cathode electrolyte (Supplementary Fig. 30b), confirming that the formate cannot be reduced at the cathode side. In addition, to investigate whether the possible accumulation of anode products on cathode after long-term electrolysis will affect the hydrogen production, the Faraday efficiencies for H$_2$ by using a two-electrode system without membrane at varied potentials from the beginning to the end of 12 h glycerol electrolysis were also calculated to be close to 100% (Fig. 7f), indicating an extremely high efficiency of electron transfer from H$_2$O to H$_2$ during the overall electrolyzing process. All the above results demonstrate that the produced formate cannot be reduced at the cathode side which will contribute to no migrational current. In conclusion, the established membrane-free electrolytic approach with Ni–Mo–N/CFC as both anode and cathode electro-catalysts and 0.1 M glycerol solution in 1 M KOH as electrolyte, is a low-cost and highly efficient strategy for the simultaneous electrochemical hydrogen and formate productions.

In summary, a strategy for the concurrent and highly efficient electrolytic productions of H$_2$ and value-added formate in the alkaline electrolyte of glycerol has been developed by utilizing Ni–Mo–N/CFC electro-catalyst as both anodic and cathodic catalysts. The obtained catalyst exhibits outstanding electro-catalytic performance for glycerol oxidation at the anode, offering

the current density of 10 mA cm$^{-2}$ at as low as 1.30 V. Owing to the much more favorable thermodynamics of glycerol oxidation OER, the two-electrode configuration only requires a quite low cell voltage of 1.36 V at 10 mA cm$^{-2}$ for H$_2$ production in the alkaline glycerol solution, which is 260 mV lower than that of conventional overall water splitting under the same catalysis by Ni–Mo–N/CFC at both anode and cathode. In addition, the ultrahigh selective electro-oxidation conversion of glycerol into value-added formate with high yield has been demonstrated. The Faraday efficiencies for H$_2$ and formate productions are 99.7% and 95.0% at cathode and anode, respectively. The high catalytic activity, as well as the long-term stability of the catalyst were also demonstrated. More significantly, it has been confirmed that the formation of formate is a consequence of successive-stepped reaction during a long-time electrolysis and the formed formate comes from both the secondary and primary carbons of glycerol. This work provides a non-noble metal Ni–Mo–N/CFC electro-catalyst for the highly efficient, energy-saving and low-cost productions of valuable H$_2$ and formate.

## Methods
**Chemicals**. Ni(CH$_3$COO)$_2$·4H$_2$O (99.0%, AR, grade), potassium formate (98%, AR, grade) and potassium hydroxide was bought from Macklin. (NH$_4$)$_6$Mo$_7$O$_{24}$·4H$_2$O (99.0%, RG, grade), maleic acid (99%, RG, grade) and deuterated water (D$_2$O) was bought from Adamas Reagent Co., Ltd. (Tansoole). Polyvinylpyrrolidone (PVP) was bought from general reagent. Glycerol (AR) was bought from Chinese medicine reagent. 2-$^{13}$C and 1, 3-$^{13}$C glycerol were purchased from Guangzhou Puen. All chemicals were used as received without any further purification. Deionized water (DIW) was used in all experiments.

**Material synthesis**. To synthesize the Ni–Mo–N/CFC electrocatalyst, NiMo-precursor was first constructed on carbon fiber cloth (NiMo-pre/CFC) through a

hydrothermal reaction. First, the CFC (5 cm × 4 cm × 0.32 mm) was pretreated in $H_2SO_4$ and $HNO_3$ mixed solution and washed with DIW by ultrasonication. Second, Ni $(CH_3COO)_2\cdot4H_2O$ (0.7475 g), $(NH_4)_6Mo_7O_{24}\cdot4H_2O$ (0.5319 g), and PVP (0.2538 g) were successively dispersed in DIW (80 mL) under stirring for 40 min. The mixture (75 ml) was transferred into a 100 mL Teflon-lined stainless steel autoclave (ANHUI CHEM$^N$, HT-100H-316L). Then the CFC was immersed into the solution and placed against the wall of autoclave. Third, the autoclave was maintained at 180 °C for 15 h. The CFC with green precipitates on the surface was taken out and washed with DIW to remove any unreacted residues. The NiMo-pre/CFC was dried at 60 °C for overnight. Finally, the as-constructed NiMo-pre/CFC was heated at 500 °C for 2.5 h in a $NH_3$ atmosphere, and then, the Ni–Mo–N/CFC electrocatalyst anchored on the CFC was obtained. The mass loading of Ni–Mo–N/CFC on CFC is ≈2.9 mg cm$^{-2}$, which is obtained by dissolving the material in nitric acid. The Ni–Mo–N/CFC-400 and Ni–Mo–N/CFC-600 were fabricated after the NiMo-pre/CFCs were annealed in an $NH_3$ atmosphere at 400 and 600 °C, respectively, for 2.5 h. The pure Ni nanoparticles on CFC was also synthesized by changing precursor and calcination atmosphere ($(NH_4)_6Mo_7O_{24}\cdot4H_2O$ is replaced by urea in precursor synthesis and $NH_3$ by $H_2/Ar$ in calcination atmosphere).

**Materials characterization**. XRD measurements were carried out on a Rigaku D/MAX 2550 diffractometer with Cu Ka radiation ($l = 1.5418$ Å). Scanning electron microscope (SEM) images were acquired on a Hitachi S-4800 SEM operating at an accelerating voltage of 3 kV. TEM images were acquired on a JEM-2100F equipped with operated at 200 kV. The chemical composition was obtained by X-ray photoelectron spectroscopy (XPS, Thermo ESCALAB 250Xi) with Mg Kα radiation source ($hv = 1253.6$ eV). The position of the C 1 s peak at 284.6 eV was employed to be a calibration reference to determine the accurate binding energies (±0.1 eV). The deconvolution of the Ni, Mo, and N was performed through a software XPSPEAK version 4.1. ICP-OES analysis was recorded on Agilent 700 Series instrument.

**Electrochemical measurements**. All measurements for HER, OER, and glycerol oxidation were conducted on a BioLogic VSP-300 electrochemical workstation in a three-electrode cell at room temperature. Carbon rod and Ag/AgCl (sat. KCl) were used as the counter and reference electrodes, respectively. The potentials measured were calibrated with respect to reversible hydrogen electrode (RHE) according to the following equation: $E_{RHE} = E_{Ag/AgCl} + 0.059$ pH $+ 0.195$. Commercial Pt/C (20 wt%, Shanghai Hesen Electric Co., Ltd.) and $IrO_2$ (99.9 wt%, Adamas-beta) were loaded on CFC with the mass loading of 2 mg cm$^{-2}$ as a contrast. For two-electrode configuration, Ni–Mo–N/CFC was used as both anode and cathode. EIS measurements were performed in a frequency range from 100,000 to 0.1 Hz with 5 mV amplitude. The electrochemical double layer capacitances (Cdl) of various samples were confirmed by CV in the potential region without faradaic process to calculate the ECSA. LSVs were measured at a scan rate of 2 mV s$^{-1}$. All the curves were used without IR compensation. All the electrochemical measurements were in $N_2$-saturated electrolyte.

**Product quantification**. The chronoamperometry testing at 1.35 V vs. RHE (for three-electrode system) or at the cell voltage of 1.4 V (for two-electrode system) were carried out to determine the products of glycerol oxidation and calculate the corresponding Faradaic efficiencies. Five millilitre of electrolyte solution were collected after approximately 12 h and then analyzed by nuclear magnetic resonance (NMR) spectrometer and ion chromatograph. $^1H$ and $^{13}C$ NMR spectra were recorded on an Avance II 300 instruments (Bruker). In which 500 μL electrolyte was added with 100 μL $D_2O$, Maleic acid was used as an internal standard. The ion chromatograph (ICS-2500, DIONEX) was equipped with a Dionex Ion-Pac$^{TM}$ AS18 (2 × 250 mm) anion separation column and the eluent was 0.1 M NaOH. The standard solutions of the HCOOK were examined under the same conditions. The relative amounts of reactant and products were calculated as $n/0.3$ mol × 100% ($n$ is the mol of carbon atom in reactant or products). Meanwhile, $H_2$ produced at the cathode was measured by a gas chromatograph (Ramiin GC2060) equipped with a packed column and a thermal conductivity detector, quantified by the external standard method. The standard curve of $H_2$ is exhibited in Supplementary Fig. 31. For each measurement, samples of 0.3 mL were collected from the same position of sealed cell and injected into the GC instrument carefully to determine the amount of $H_2$ produced. The theoretical generated $H_2$ amount was calculated as $Q_{tot} × V_m/(Z × F)$ ($Q_{tot}$ is the total charge passed through the electrodes, $V_m$ is the molar volume of gas, $Z = 2$ is the number of electrons needed to produce a molecule of $H_2$, $F = 96485$ C mol$^{-1}$ is the Faraday constant). As shown in Fig. 7e, when 12.09 C charge calculated by electrochemical workstation passed through the electrochemical cell, 1.506 mL $H_2$ measured by GC was generated on the cathode, which is in nearly accordance with the theoretical value of 1.516 mL. All products quantification was obtained after 20 cycles of CV scans and 3 LSV tests, when the substantial and slight dissolution of Mo species respectively at anode and cathode have almost ended.

The yield (%) and selectivity (%) of formate formation can be determined by the following Eqs. (1) and (2), respectively

$$Yield(\%) = \frac{N(\text{formate yield})}{3 × N(\text{initial glycerol})} × 100\%, \tag{1}$$

$$Selectivity(\%) = \frac{N(\text{formate yield})}{3 × N(\text{glycerol consumed})} × 100\%. \tag{2}$$

The Faraday efficiency (%) of the formate and $H_2$ production can be determined by the following Eqs. (3) and (4), respectively

$$FE(\%) = \frac{N(\text{formate yield})}{Q_{totl}/(Z_1 × F)} × 100\%, \tag{3}$$

$$FE(\%) = \frac{N(H_2 \text{ Production})}{Q_{totl2}/(Z_2 × F)} × 100\%. \tag{4}$$

Where $Q_{tot1}$ and $Q_{tot2}$ are the total charge passed through the electrodes, $z_1 = 8/3$ is the number of electrons that form a mole of formate, $z_2 = 2$ is the number of electrons that produce a molecule of $H_2$, $F$ is the Faraday constant (96,485 C mol$^{-1}$).

**Isotope labeled study of glycerol**. Four millilitre of 1 M KOH solution containing 0.1 M of $^{13}C$-labeled glycerol (1, 3-$^{13}C$ or 2-$^{13}C$) as electrolyte was employed to participate in the glycerol electro-oxidation reaction at 1.35 V vs. RHE with Ni–Mo–N/CFC as anode catalyst, according to the $^1H$ and $^{13}C$ NMR spectra analyses of the production from electrolytes with different $^{13}C$-labeled reagent. Aliquots were sampled every 25.72 for 77.16 C mL$^{-1}$.

## Data availability
All data is available from the authors upon reasonable request.

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

## Acknowledgements
This work was supported by National Natural Science Foundation of China (51702099), Shanghai Sailing Program (17YF1403800), China Postdoctoral Science Foundation funded project (2017M611500), the Opening Project of State Key Laboratory of High Performance Ceramics and Superfine Microstructure (SKL201702SIC). We gratefully acknowledge Dr. Guirong Zhang at the *East China Normal University* for his help with the statistical analysis.

## Author contributions
J.S. and L.C. led the project. Y.L. and X.W. contributed equally to this work. Y.L. designed and performed the experiments; X.W. assisted with the experiments. M. H contributed considerably to the revision of the manuscript. All authors discussed the results and commented on the manuscript.

## Competing interests
The authors declare no competing interests.
