## [Peer Review File · Nature Communications]

Reviewers' comments:

Reviewer #1 (Remarks to the Author):

This work reports a bifunctional electrocatalyst, nickel-molybdenum-nitride loaded on carbon fiber cloth (Ni-Mo-N/CFC) and its application to the concurrent electrolytic productions of high-purity hydrogen and value-added formate in an alkaline glycerol solution. In this work, glycerol is oxidized to formate at the anode catalyzed by Ni-Mo-N/CFC, and simultaneously hydrogen is generated under the catalysis by the same catalyst of Ni-Mo-N/CFC at the cathode. This two-electrode configuration requires a relatively low cell voltage of 1.36 V at 10 mA cm⁻² in the alkaline glycerol solution, which is 260 mV lower than that of conventional overall water splitting system employed by the Ni-Mo-N/CFC at both anode and cathode. More importantly, the Faradaic efficiency for hydrogen and formate productions was close to 100 % and 95 % at cathode and anode, respectively. The paper is well written and logically organized in the aspects of the correlation between nature of electrocatalysts, reaction conditions, and activity/selectivity in electrochemical devices. Given the importance and growing interests of high purity hydrogen production and electrocatalytic conversion of biomass-derived chemicals to value-added products, this work is meritorious and timely. I recommend the publication of this paper in the Nature Communications after addressing the following points.

[1] As examined by ICP-OES analysis, the Ni-Mo-N/CFC was found to lose more than 90% of molybdenum after the glycerol oxidation at the anode, either in a short-time or a long-time electrochemical test. On the other hand, only a small amount of Mo was dissolved from the Ni-Mo-N/CFC catalyst at the cathode for the HER. The reasonable explanation should be addressed for this different behavior.

[2] The authors carried out surface analysis of Ni-Mo-N/CFC after the glycerol oxidation and HER. However, surface analysis in ex-situ manners could give limitations to estimate the oxidation states of the Ni-Mo-N/CFC during the real operational condition. Since the surface state can be interfered during the sample storage and transfer, the results of surface analysis might be of an indirect observation of the catalyst surface, which would be not sufficient to elucidate the origin of the enhanced activity for glycerol oxidation and HER. Discussion is needed.

[3] Some more reports for the electrooxidation of biomass-derived glycerol into value-added chemicals, such as [ChemCatChem 2017, 9, 1683-1690; Green Chem. 2016, 18, 2877-2887; ChemSusChem 2014, 7, 1051-1056; Appl. Catal. B-Environ. 2019, 245, 555-568], should be included and stated in order to strengthen the literature works in this paper.

[4] The 'Discussion' section looks like 'Conclusions'. The authors need to comprehensively discuss on their study and new findings if the results and discussion can be split in this manuscript.

Reviewer #2 (Remarks to the Author):

The manuscript proposed by Li et al reported the synthesis of a non-noble complex material applied as cathode and anode for an all-alkaline membraneless glycerol electrolyzer. The use of same electrode material with the same loading (it seems it is) is intriguing, since a better efficiency could be achieved whether the authors use another cathode. However, the use of exact same material save time in practical applications, since the cell may be designed to exchange cathode and anode during continuous use, which reactivates the electrodes. Thus, the Ni-Mo-N/CFC can be called a smart material.

In general this work is remarkable in terms of characterization of an applied catalyst, since the synthesis to the stability investigation. However, there are fundamental and practical issues that must

be addressed in order to make it suitable to be reevaluated as a MS for Nature Communications.

The main issue is the lack of information regarding the glycerol electrolysis. I could not identify the exact experiment. My point is, a linear sweep potential is not glycerol electrolysis from the practical point of view and the authors know of that for sure, since the investigation of the stability is clearly a practical application (Fig. 3h). On the other hand, from a fundamental investigation, the work also lacks information on the half-cell measurements. Thus, here are my comments:

- To fundament and justify their work, the authors state that there is no need for membrane in glycerol electrolysis to produce high market-price compounds and clean hydrogen. That is not true, because it depends on the time of electrolysis and the hydrodynamics. For a stationary cell in a long-time glycerol electrolysis, the non-protonated oxidation product formed at the anode may be reduced at the cathode. Furthermore, the accumulation of side product on the cathode may create an internal potential difference, which may lead to migrational current contribution. Since the glycerol electrolysis is not clear in the text, it seems we cannot disregard any of these hypothesis.

- Regarding half-cell measurements, the authors obtained important electrocatalytic information from LSV. Firstly, I would like to see any comments on the origin of the anodic current at around 1.4 V in the glycerol electrooxidation experiments. Why it appears only at specific cases? Furthermore, the authors must state whether the LSV is representative (or stable), since the first LSV for glycerol electrooxidation (even in alkaline medium) is not representative. The first dissociative adsorption (at the first LSV) may not match the second onwards, which may not be the case, but we cannot take such conclusion from the text. Thus, the authors may either state that or show 3 successive LSVs in the SI for example.

- The electrolysis of glycerol must be detailed. Is it a long-time electrolysis? Do the authors collect the sample for chromatography and NMR after 12 h under applied potential equivalent to 10 mA cm⁻²? How is the sample collected? If it is a long-time electrolysis, how do the authors avoid migrational current using three-electrode conventional cell? How do the authors guarantee that carboxylate compounds are not reduced at the cathode side? All these questions must be addressed or they could be ignored, it depends on how the experiment was performed.

- Formate is an obvious product after long-time electrolysis. It is a consequence of successive stepped reaction, leading to high oxidation state. However, the authors found only formate as a product, which has been found using noble-metal catalyst. The authors could comment on that.

- The authors state that carbonate found by NMR is a consequence of CO₂ from air going through the alkaline medium. However, the applied potential is too high... high enough to lead the reaction to formate. How do the authors decouple carbonate formation from the electrochemical surface reaction and from air?

- To detail the pathway of the reaction, which the authors called mechanism, they fundament it on the reference [46], which is inadequate. Reference [46] deals with heterogeneous organic reaction which has nothing to do with surface electrochemical reaction. The author must use an appropriate reference. There are dozens of papers, some chapters and some reviews regarding the paths of such reaction in alkaline medium. Whether the authors want to justify their working by showing a paper in which formate has been found from glycerol electrooxidation in alkaline medium, they may find it, specially on Pd surface. The authors must revisit the papers of electrocatalysis.

- The authors also may like to review some minor errors and typos throughout the text.

Reviewer #3 (Remarks to the Author):

The manuscript presented the synthesis and application of Ni-Mo-nitride heterostructure electrocatalyst supported on carbon fiber cloth for concurrent volarization of glycerol aqueous solution. The highly selective production of formate and efficient hydrogen evolution at relatively low overpotential were observed. It was claimed the catalyst at anode and cathode is stable with exceptionally high efficiency. The research idea of electrolytic volarisation biomass-based wastes is interesting yet significant defects exist in the current manuscript, so it is not suitable to be accepted.

The issues existing in the manuscript are listed below:

- 1.) The state of the arts of reducing overpotentials in electrolysis is not well summarised and highlighted, in particular for those catalysts with high faradic efficiency and stability. Although it claimed "no reports can be found on such (bifunctional) electrocatalysts", there are a few already cited as listed in references of this manuscript.
- 2.) Formates were stated value-added products relative to glycerol, though it was not well justified enough how significant is the added value. The value of the formate formed in electrolyte solution seems not competitive enough compared to the raw material, even some wastes because of the barrier in separation.
- 3.) Altraselective production of formate was stated in the work, though the reason for the ultrahigh selectivity was not argued and justified from the thermodynamic and kinetic points of view. This makes the scientific contributions of the work rather weak.
- 4.) A few problems of the characterisation results: The statement of "nanosheet" is not convincing from the TEM and SEM results. The thickness is not measured and the diameter stated is not accurate as comparing SEM and TEM images. The EDAX and XPS results of Carbon and Oxygen should have been presented though they are missing. The carbon fibre cloth should also been tested and compared with the presented catalysts because the NH₃ annealing could have already nitridised the CFC. The peaks in the LSV results in Fig 3a, Fig 5 and supporting materials were not discussed. It seems the anodic catalysts have been oxidised.
- 5.) many typos and grammatic errors should be corrected. For example, " in the Mo 3p-N 1s spectrum..." is very confusing.
- 6.) There are no potential changes in stability tests for anode and cathode, and it claimed the catalysts are stable in use. These claims are not convincing enough because significant Mo has lost as measure for the post-reaction catalysts.
- 7.) Mecahnism was proposed though the elementary and global reactions are missing. The kinetic control and thermodynamics of the related reactions should have been discussed, otherwise it is not foundational for the selectivity of the work.
- 8.) A few glycerol concentration was tested in electrolysis, what about lower concentration? would the lower concentration affect the stability? These experiments would benefit mechanism discussion.
- 9.) The XPS analyses of Ni species are not convincing, in particular the Fig. 8d, why no Ni (III) and less Ni (0) were observed there? They Ni species amount can be quantified. Catalyst Vacancies were claimed to form on the electrode, though they were not proved.
- 10.) It stated Ni (0) on cathod was converted to Ni (II) in reaction, how could they in reducing conditions?
- 11.) 90% Mo lossing was determined, why the materials still maintained morphology unchanged? why could they be stable in electrolysis? The discussion and results are not convincing, the long-term operation plausibility is dubious as well.
- 12.) Discussions were presented though it only repeated abstract and no critical discussions were presented, supposing thwe work is not completed.
- 13.) The experimental is not detailed enough and description is vague. for example the conductivity or resistance of CFC, source etc. The NH₃ treatments at 400 and 600 degC are confusing too. The ESCA and CdI calculations should have been presented.

Hence, I would like to suggest rejecting the manuscript.

Response to Reviewer 1

Comments and suggestions from Reviewer 1.

This work reports a bifunctional electrocatalyst, nickel-molybdenum-nitride loaded on carbon fiber cloth (Ni-Mo-N/CFC) and its application to the concurrent electrolytic productions of high-purity hydrogen and value-added formate in an alkaline glycerol solution. In this work, glycerol is oxidized to formate at the anode catalyzed by Ni-Mo-N/CFC, and simultaneously hydrogen is generated under the catalysis by the same catalyst of Ni-Mo-N/CFC at the cathode. This two-electrode configuration requires a relatively low cell voltage of 1.36 V at 10 mA cm⁻² in the alkaline glycerol solution, which is 260 mV lower than that of conventional overall water splitting system employed by the Ni-Mo-N/CFC at both anode and cathode. More importantly, the Faradaic efficiency for hydrogen and formate productions was close to 100 % and 95 % at cathode and anode, respectively. The paper is well written and logically organized in the aspects of the correlation between nature of electrocatalysts, reaction conditions, and activity/selectivity in electrochemical devices. Given the importance and growing interests of high purity hydrogen production and electrocatalytic conversion of biomass-derived chemicals to value-added products, this work is meritorious and timely. I recommend the publication of this paper in the Nature Communications after addressing the following points.

Response: Thank you very much for the positive comment and recommendation. Please find the following detailed responses to your comments and suggestions.

1. As examined by ICP-OES analysis, the Ni-Mo-N/CFC was found to lose more than 90% of molybdenum after the glycerol oxidation at the anode, either in a short-time or a long-time electrochemical test. On the other hand, only a small amount of Mo was dissolved from the Ni-Mo-N/CFC catalyst at the cathode for the HER. The reasonable

explanation should be addressed for this different behavior.

Response: Thank you for the question. In this work, Ni-Mo-N/CFC was found to lose 90% of molybdenum after the glycerol oxidation at the anode, whereas only a small amount of Mo was dissolved at the cathode for the HER. Such a difference results from the fact that Mo^{6+} species can be well dissolved in alkaline solution in the form of molybdate and polymolybdate. Almost all Mo species can be oxidized to Mo^{6+} in oxidizing conditions as shown in the high-resolution Mo 3d XPS spectra of post-Gly Ni-Mo-N/CFC (**Supplementary Fig. 27a**), leading to a majority of Mo being leached into the electrolyte. However, under the reduction conditions, a minor amount of Mo species were converted to Mo^{6+} species due to the exposure to air, which means a limited amount of Mo loss at the cathode. These information has been added in the revised manuscript (Page 22).

2. The authors carried out surface analysis of Ni-Mo-N/CFC after the glycerol oxidation and HER. However, surface analysis in ex-situ manners could give limitations to estimate the oxidation states of the Ni-Mo-N/CFC during the real operational condition. Since the surface state can be interfered during the sample storage and transfer, the results of surface analysis might be of an indirect observation of the catalyst surface, which would be not sufficient to elucidate the origin of the enhanced activity for glycerol oxidation and HER. Discussion is needed.

Response: Many thanks for your kind suggestion. Yes, ex-situ surface analysis suffers from the limitations in estimating the real origin of catalytic activity. To get more precise results, samples were carefully handled before and during these analysis by, for example, isolating these samples from oxygen, shortening the storage and transfer time intervals, etc. We haven't had enough time to carry out in-situ analyses of samples at the current stage due to the limited time period of manuscript revision. We would like to do more in-situ analysis in the following researches.

3. Some more reports for the electrooxidation of biomass-derived glycerol into value-added chemicals, such as [ChemCatChem 2017, 9, 1683-1690; Green Chem.

2016, 18, 2877-2887; ChemSusChem 2014, 7, 1051-1056; Appl. Catal. B-Environ. 2019, 245, 555-568], should be included and stated in order to strengthen the literature works in this paper.

Response: Thank you for your kind comment. These references are critically important and closely related to this manuscript, therefore have been cited in the revised manuscript (Ref. 25, 26, 54 and 57).

4. The ‘Discussion’ section looks like ‘Conclusions’. The authors need to comprehensively discuss on their study and new findings if the results and discussion can be split in this manuscript.

Response: Thank you for your constructive comments and suggestions. We have carefully revised the structure of the manuscript and added much discussion on our findings to make logic clearer.

Response to Reviewer 2

Comments and suggestions from Reviewer 2.

The manuscript proposed by Li et al reported the synthesis of a non-noble complex material applied as cathode and anode for an all-alkaline membraneless glycerol electrolyzer. The use of same electrode material with the same loading (it seems it is) is intriguing, since a better efficiency could be achieved whether the authors use another cathode. However, the use of exact same material save time in practical applications, since the cell may be designed to exchange cathode and anode during cotinuous use, which reactivates the electrodes. Thus, the Ni-Mo-N/CFC can be called a smart material.

In general this work is remarkable in terms of chacarterization of an applied catalyst, since the synthesis to the stability investigation. However, there are fundamental and practical issues that must be addressed in order to make it suitable to be reevaluated as a MS for Nature Communications.

The main issue is the lack of information regarding the glycerol electrolysis. I could not identify the exact experiment. My point is, a linear sweep potential is not glycerol electrolysis from the practical point of view and the authors know of that for sure, since the investigation of the stability is clearly a practical application (Fig. 3h). On the other hand, from a fundamental investigation, the work also lacks information on the half-cell measurements. Thus, here are my comments:

Response: Thank you very much for the kind comments and suggestions. Please find the following detailed responses.

1. To fundament and justify their work, the authors state that there is no need for membrane in glycerol electrolysis to produce high market-price compounds and clean hydrogen. That is not true, because it depends on the time of electrolysis and the hydrodynamics. For a stationary cell in a long-time glycerol electrolysis, the

non-protonated oxidation product formed at the anode may be reduced at the cathode. Furthermore, the accumulation of side product on the cathode may create an internal potential difference, which may lead to migrational current contribution. Since the glycerol electrolysis is not clear in the text, it seems we cannot disregard any of these hypothesis.

Response: *Thank you very much for the constructive suggestions and questions. Yes, we cannot disregard any of the hypotheses such as the formation of non-protonated oxidation product and accumulation of side product on the cathode. To make the issues clearer, we further carried out the experiments to evaluate the influence of anode products on cathode. A two-electrode electrolyzer with a Nafion membrane for the concurrent electrochemical hydrogen production and formate reduction has been set up by adding 0.1 M glycerol dissolved in 1 M KOH to the anode and 0.1 M formate dissolved in 1 M KOH in cathode compartments. It has been found that the Faraday efficiency values for hydrogen production at varied current densities in this electrolyzer are consistently close to 100% (Supplementary Fig. 30a). As examined by the NMR analysis (Supplementary Fig. 30b), the products in the anode are identical to those from the membrane-free system, and only water and formate can be detected in the cathode electrolyte, demonstrating that the added formate was not reduced at the cathode. In addition, the Faraday efficiency for hydrogen by using a two-electrode system without using membrane is also close to 100 % at varied potentials from the beginning to the end of 12 h glycerol electrolysis (Figure 7e), indicating an extremely high efficiency of electron transfer from H₂O to H₂ during the overall electrolysis process. Nevertheless, in spite of the ~100% Faraday efficiency for hydrogen in both systems even after a long time glycerol electrolysis and no other products can be detected except formate and carbonate, the presents of trace amounts of some non-detectable non-protonated oxidation products cannot be totally excluded, which, if present, may still lead to migrational current contribution. However, under the current experiment conditions, it is confident that the formate is the only major product of glycerol electro-oxidation, and the membrane is unnecessary in the present*

case not only based on the above reasons, but also because the use of membrane will lead to much increased cell cost and the resistance for species diffusions.

2. Regarding half-cell measurements, the authors obtained important electrocatalytic information from LSV. Firstly, I would like to see any comments on the origin of the anodic current at around 1.4 V in the glycerol electrooxidation experiments. Why it appears only at specific cases? Furthermore, the authors must state whether the LSV is representative (or stable), since the first LSV for glycerol electrooxidation (even in alkaline medium) is not representative. The first dissociative adsorption (at the first LSV) may not match the second onwards, which may not be the case, but we cannot take such conclusion from the text. Thus, the authors may either state that or show 3 successive LSVs in the SI for example.

Response: *Thank you for the constructive suggestion. The C-C bond cannot be completely broken down at relatively low potentials and oxygen evolution reaction will take place at reasonably elevated potentials. So, the potentials of 1.35 V in the half-cell measurements and 1.4 V in the two-electrode system were selected as the compromised potential values without any other special comments (Supplementary Fig. 15). This statement has been added in the revised manuscript. In addition, the LSV for glycerol electrooxidation is representative because all the LSV curves were obtained after 20 cycles of cyclical voltammetry and 3 LSV tests for stabilizing the current. According to the reviewer's suggestion, the 3 successive LSVs have been shown in the SI (Supplementary Fig. 7).*

3. The electrolysis of glycerol must be detailed. Is it a long-time electrolysis? Do the authors collect the sample for chromatography and NMR after 12 h under applied potential equivalent to 10 mA cm⁻²? How is the sample collected? If it is a long-time electrolysis, how do the authors avoid migrational current using three-electrode conventional cell? How do the authors guarantee that carboxylate compounds are not reduced at the cathode side? All these questions must be addressed or they could be ignored, it depends on how the experiment was performed.

Response: Thank you very much for the constructive suggestions and questions. Experimental details on glycerol electrolysis has been provided in the revised manuscript according to the reviewer's kind suggestion (main text and Methods). Yes, the electrolysis of glycerol is a long-time electrolysis. Reaction has been performed in 5 ml 1 M KOH solution containing 0.1 M glycerol at room temperature at 1.35 V equivalent to 25 mA cm⁻² and 1.4 V equivalent to 15 mA cm⁻² for 12 h, after that the electrolyte samples were collected using a transfer pipette after the electrolysis for chromatography and NMR analyses. To investigate whether the possible accumulation of anode products on cathode after long-term electrolysis will affect the hydrogen production, the Faraday efficiency for hydrogen production as determined by a conventional two-electrode cell without using a membrane is consistently close to 100 % at varied potentials from the beginning to the end of 12 h glycerol electrolysis process (**Figure 7e**), indicating an extremely high efficiency of electron transfer from H₂O to H₂ during the overall electrolysis. Formate is the only carboxylate product detected in this system. To evaluate the possible formate reduction at the cathode, a three-electrode configuration for HER has been set up by adding 0.1 M formate in 1 M KOH electrolyte. No formate reduction can be detected as examined by NMR (**Supplementary Fig. 18**). A two-electrode electrolyzer with a Nafion membrane for the concurrent electrochemical hydrogen and formate productions has been set up by adding 0.1 M glycerol dissolved in 1 M KOH to the anode and 0.1 M formate dissolved in 1 M KOH in the cathode compartments. It has been found that the Faraday efficiencies for hydrogen at different current densities are still consistently close to 100% (**Supplementary Fig. 30a**). Only water and formate can be detected in the cathode electrolyte, as examined by NMR analysis (**Supplementary Fig. 30b**). These results indicate that the produced formate cannot be reduced at the cathode side and therefore have no contribution to migrational current under the current conditions. Non-protonated oxidation products, which may lead to migrational current contribution, if present, cannot be detected in our experiments, indicating the negligible contributions of these non-protonated products to migrational current, even though the presence of trace amount of these products

cannot be totally excluded. Theoretically, to absolutely avoid the migrational current using three-electrode conventional cell, the membrane is needed between the cathode and the anode as you suggested, however, very fortunately, the membrane seems unnecessary in this case.

4. Formate is an obvious product after long-time electrolysis. It is a consequence of successive stepped reaction, leading to high oxidation state. However, the authors found only formate as a product, which has been found using noble-metal catalyst. The authors could comment on that.

Response: *Thank you for your kind comment. However, at present, we cannot find any report on the formate-only production using noble-metal catalyst, which, therefore need further verification. We have pointed out this issue in our revised manuscript.*

5. The authors state that carbonate found by NMR is a consequence of CO₂ from air going through the alkaline medium. However, the applied potential is too high... high enough to lead the reaction to formate. How do the authors decoupled carbonate formation from the electrochemical surface reaction and from air?

Response: *Thank you very much for the constructive questions. According to your suggestion, we further detected carbonate formation by ion chromatography to explore the origin of carbonate. Electrolytes with and without glycerol electro-oxidation have been examined. It has been found that carbonates came not only from air but also from the electrochemical reaction. However, noticeably, the Faradic efficiency for carbonate is as low as 2.4%, which is a very small value indeed (the high solubility of CO₂ in alkaline medium may lead to some errors although efforts have been made to control the experimental conditions). Although the carbonates from electrochemical surface reaction is extremely few compared to the carbonates from air, our expression in previous manuscript may not be accurate enough. Thank you again for your valuable advice, and we have modified the description in the article to make it more accurate and clear.*

6. To detail the pathway of the reaction, which the authors called mechanism, they fundament it on the reference [46], which is inadequate. Reference [46] deals with heterogeneous organic reaction which has nothing to do with surface electrochemical reaction. The author must use an appropriate reference. There are dozens of papers, some chapters and some reviews regarding the paths of such reaction in alkaline medium. Whether the authors want to justify their working by showing a paper in which formate has been found from glycerol electrooxidation in alkaline medium, they may find it, specially on Pd surface. The authors must revisit the papers of electrocatalysis.

***Response:** Thank you very much for the constructive comment. According to the suggestion, we have added experiments and discussed the experimental results related to the pathway of the reaction in the revised manuscript. A large number of papers about glycerol electro-oxidation in alkaline medium, especially on Pd surface have also been revisited. **Figure 3g** shows the ^1H NMR results of glycerol and product (formate) during the whole glycerol electrolysis period at 1.35 V vs RHE. The decrease of glycerol amount and the enhancement of formate concentration in the time course of electrocatalysis can be clearly observed, undoubtedly indicating the conversion of glycerol to formate. During the process of a magnitude of electric charge of ~ 385 C passing through the electrochemical cell, the concentration of formate increased to the maximum and that of glycerol decreased to 0, which suggests the complete conversion of glycerol, leading to a yield of 93% for formate production **(Figure 3h)**. Surprisingly, methanol was undoubtedly detected and the ratio of methanol to formate increased to around 1:5 at ~ 129 C charges transferred, and then began to decrease until it cannot be detected. To explore the source of methanol, a three-electrode configuration using Ni-Mo-N/CFC catalyst was set up by adding 0.1 M formate dissolved in 1 M KOH at the cathode. No other products were detected except hydrogen at the cathode side **(Supplementary Fig. 18)**, confirming that methanol does not come from formate reduction of cathodic reaction. Fortunately, formaldehyde was detected via phloroglucinol method, a highly sensitive method for*

detecting formaldehyde with a detection limit of 0.1ppm (Supplementary Fig. 19). So, we infer that the methanol did come from the Cannizzaro reaction (an aldehyde without an α -hydrogen atom undergoes an intermolecular redox reaction under the action of a strong base to form a carboxylic acid and an alcohol) of formaldehyde in alkaline solution. Besides, we also found that the carbon atoms in methanol molecules came from carbon atoms at positions 1, 3 of glycerol according to the results of isotope tracer described below.

To better understand the mechanism of glycerol electro-oxidation to formate, experiments using 0.1 M 1, 3- ^{13}C -labeled glycerol and 0.1 M 2- ^{13}C -labeled glycerol in 1M KOH as electrolyte and Ni-Mo-N/CFC as electrocatalyst have been further performed at 1.35 V vs RHE (figure 4). From the ^1H NMR analysis it can be found that the ratios of unlabeled formate to ^{13}C -labeled formate obtained by 2- ^{13}C -labeled glycerol oxidation are 0.72:1, 1.39:1, 1.9:1, and the corresponding ratios obtained by 1,3- ^{13}C -labeled glycerol oxidation are 1:0.82, 1:1.42, 1:1.98, respectively, during the progress of the glycerol oxidation reaction to varied stages, as determined by ^1H NMR analysis. Therefore, it is clear that the formed product, formate, comes from both the secondary and primary carbons of glycerol.

Moreover, the methanol can be detected in 1,3- ^{13}C -labeled glycerol, while it cannot be detected in 2- ^{13}C -labeled glycerol by the ^{13}C NMR spectroscopy in monitoring the source of methanol (Supplementary Fig. 20), which means that the formation of formate is a consequence of successively stepped reaction after a long-time electrolysis with methanol being one of the major intermediates, as described and supposed in more detail in the following.

According to the above results and previous reports (Ref. 24, 30, 52-57), a possible reaction path of glycerol electro-oxidation at the anode to formate is proposed as shown in Scheme 1. Firstly, the formation of formate begins with glycerol oxidation to glyceraldehyde, which is then oxidized to formate and glycolaldehyde with the breakage of the C-C bonds. Next, the oxidative cleavage of the glycolaldehyde produces formate and formaldehyde, followed by the methanol and

formate formations by the intermolecular redox reactions (Cannizzaro reaction) of formaldehyde in alkaline solution, and the methanol final oxidation to formate. In all, almost all of the reactant glycerol and several intermediates are eventually oxidized to formate. In this overall pathway of glycerol to formate, a very small amount of carbonate (~2.4%) may come from the further oxidation of formate.

In addition, weak peaks of glycollic acid can be detected in the ^{13}C -labeled glycerol oxidation products by the ^{13}C NMR spectroscopy due to the labeling of ^{13}C though they cannot be detected in the unlabeled glycerol oxidation products, indicating that the successive oxidation of glycerol to glycolic acid, and then to formate, a typical pathway of the glycerol electro-oxidation to formate, is a minor side reaction pathway in the present study.

These results and discussions have been added in the revised manuscript (Page 14-18).

7. The authors also may like to review some minor errors and typos throughout the text.

Response: *Thank you very much for the comment. We have carefully checked the whole manuscript and supporting information and the errors and typos have been corrected in the revised manuscript.*

Response to Reviewer 3

Comments and suggestions from Reviewer 3.

The manuscript presented the synthesis and application of Ni-Mo-nitride heterostructure electrocatalyst supported on carbon fiber cloth for concurrent volarization of glycerol aqueous solution. The highly selective production of formate and efficient hydrogen evolution at relatively low overpotential were observed. It was claimed the catalyst at anode and cathode is stable with exceptionally high efficiency. The research idea of electrolytic volarisation biomass-based wastes is interesting yet significant defects exist in the current manuscript, so it is not suitable to be accepted.

Response: Thank you very much for the kind comments and suggestions. Please find the following detailed responses.

1. The state of the arts of reducing overpotentials in electrolysis is not well summarised and highlighted, in particular for those catalysts with high faradic efficiency and stability. Although it claimed "no reports can be found on such (bifunctional) electrocatalysts", there are a few already cited as listed in references of this manuscript.

Response: Thank you very much for the constructive suggestions and questions. According to the reviewer's suggestion, the technical status of reducing overpotential in electrolysis has been provided in **Supplementary Table 5**. Such a low cell voltage

of the Ni-Mo-N/CFC for electro-chemical-assisted water splitting is much lower than those of most reported systems, except for several strong reducing agents such as hydrazine for assisted water splitting. For the second question, up to now, no reports can be found on such a low-cost, high performance and stable non-noble-metal bifunctional electrocatalyst which could efficiently work for the concurrent electro-catalytic glycerol oxidation and HER. A number of literatures cited in this manuscript reported noble metal materials used as catalysts for the electro-oxidation of glycerol (For example, Ref. 24-26), and few of them used non-noble metal catalysts only for the anode glycerol oxidation with rather low Faraday efficiency and selectivity at a quite high oxidation potential. (For example, Ref. 32, 50 and 59)

2. Formates were stated value-added products relative to glycerol, though it was not well justified enough how significant is the added value. The value of the formate formed in electrolyte solution seems not competitive enough compared to the raw material, even some wastes because of the barrier in separation.

Response: *Thank you very much for the constructive comment. Formate (or formic acid) is an important chemical intermediate in many industrial processes. It can be used as fuel for direct formate fuel cells and for hydrogen storage thanks to its relatively high capacity (4.4 wt%). Glycerol is a byproduct during the production of biodiesel (about 10 wt% of the total products) and are presently sold approximately at US \$110–990 / ton (Ref. 30 and 31), significantly lower than that of formic acid (approximately \$ 1300/ton (Ref. 29)), indicating the high-added value of formate. Furthermore, the electro-oxidized product of glycerol, formic acid, is much more valuable than oxygen produced by pure OER at anode, which will mix together with H₂ produced on the cathode to form an explosive H₂/O₂ gaseous mixture. These discussions have been added in the revised manuscript (Page 4).*

3. Altraselective production of formate was stated in the work, though the reason for the ultrahigh selectivity was not argued and justified from the thermodynamic and kinetic points of view. This makes the scientific contributions of the work rather

weak.

Response: Thank you very much for the constructive suggestion. The Faraday efficiency and selectivity of formate have been examined at elevated potentials as shown in **Supplementary Fig. 15**. A potential of ~ 1.35 V was determined as the optimum potential, resulting in both high selectivity ($\sim 92.48\%$) and Faradaic efficiency ($\sim 97\%$). In addition, The Gibbs free energy of the glycerol to formate oxidation is -533.0 KJ/mol, and a theoretical oxidation potential of 0.69 V (vs the standard hydrogen electrode; **Supplementary Table 1**) is required, which is far lower than that of 1.23 V for OER under the standard conditions, i.e., the glycerol electro-oxidation is thermodynamically much more favorable than OER, the latter is the rate-determining step of the overall water electrolyzing process. The detailed kinetics is still unclear presently, but a very valuable and promising issue. This paper mainly focuses on the bifunctional electrocatalyst Ni-Mo-N/CFC and its attractive performance for both HER and glycerol oxidation. More attention will be paid to the kinetics of these reactions.

4. A few problems of the characterisation results: The statement of "nanosheet" is not convincing from the TEM and SEM results. The thickness is not measured and the diameter stated is not accurate as comparing SEM and TEM images. The EDAX and XPS results of Carbon and Oxygen should have been presented though they are missing. The carbon fibre cloth should also been tested and compared with the presented catalysts because the NH₃ annealing could have already nitridised the CFC. The peaks in the LSV results in Fig 3a, Fig 5 and supporting materials were not discussed. It seems the anodic catalysts have been oxidised.

Response: Thank you very much for the constructive suggestions and questions. Yes, the statement of "nanosheet" may not be very suitable from the TEM and SEM results. Therefore, the word 'nanoplate' is used as the substitute for 'nanosheet' in the revised manuscript. The average thickness and diameter distribution of NiMo-Pre/CFC and Ni-Mo-N/CFC nanoplates have been shown in **Figure 2b, 2c and Supplementary**

Fig.1. The NiMo-Pre/CFC nanoplates are about 677 nm in diameter and 73.0 nm in thickness. The Ni-Mo-N/CFC nanoplates are approximately 682 nm in diameter and 57.9 nm in thickness.

According to the reviewer's suggestion, the EDAX and XPS results of carbon and oxygen have been provided in the revised manuscript (**Figure 2f, Supplementary Fig. 4-5 and 28-29**) and related discussions have been added in the revised manuscript (Page 8-9 and 23-24). XPS analysis of the carbon fibre cloth (CFC) have also been performed and discussed according to the reviewer's kind suggestion (**Supplementary Fig. 6**). The C1s XPS spectrum can be deconvoluted into three main peaks with binding energies of 284.6, 285.79 and 287.8 eV assigned to C-C/C=C, C-OH and C=O, respectively, which are similar to those of Ni-Mo-N/CFC, indicating that the synthetic processes have little effect on the CFC.

In addition, the discussions about the peaks in the LSV results and supporting materials have been added in the revised manuscript. The peaks in the LSV curves centered at about 1.4 V for the OER and 1.42 V for the water splitting can be ascribed to the oxidation peak of $\text{Ni}^{2+}/\text{Ni}^{3+}$ (Ref. 49). After introducing 0.1 M glycerol, the current density increased markedly, and the anodic potential strikingly decreased to 1.30 V vs RHE at 10 mA cm^{-2} . Meanwhile, the redox couple $\text{Ni}^{2+}/\text{Ni}^{3+}$ disappeared as shown in the cyclic voltammogram of glycerol electrooxidation (**Supplementary Fig.8**) probably due to the indirect charge transfer mechanism that Ni^{2+} is oxidized to Ni^{3+} and then completely consumed in the oxidation of glycerol to form Ni^{2+} , and makes the direct reduction of Ni^{3+} to Ni^{2+} impossible (Ref. 50 and 51).

5. Many typos and grammatic errors should be corrected. For example, " in the Mo 3p-N 1s spectrum..." is very confusing.

Response: Thank you very much for the comment. The expression of "in the Mo 3p-N 1s spectrum..." is due to the partial overlapping between N 1s and Mo 3p and we have made a corresponding revision in the revised manuscript. In addition, we have carefully checked the whole manuscript and supporting information and the typos and

grammatic errors have been corrected as far as possible in the revised manuscript.

6. There are no potential changes in stability tests for anode and cathode, and it claimed the catalysts are stable in use. These claims are not convincing enough because significant Mo has lost as measure for the post-reaction catalysts.

Response: *Thank you very much for the comment. According to the ICP-OES analysis, the Ni-Mo-N/CFC was found to lose more than 90% of molybdenum after the glycerol oxidation at the anode and only a small amount of Mo was dissolved from the Ni-Mo-N/CFC catalyst at the cathode for the HER, either in a short-time (20 cycles of CV scans + 3LSV) or a long-time (20 cycles of CV scans + 3LSV + 12h CP) electrochemical test period (Supplementary Table 2 and 3). Ratios of Ni to Mo elements in anodic Ni-Mo-N/CFC catalyst after a short-time and a long-time electrochemical glycerol anodic oxidation are 1:0.15 and 1:0.13, and the corresponding concentrations of Ni and Mo elements in the electrolyte are 3.86 ppm and 3.17 ppm, respectively. The stability tests were started after 20 cycles of CV and 3 LSV tests and little molybdenum was lost during the prolonged stability tests of glycerol anodic oxidation. The catalyst with very little molybdenum amount after the initial massive Mo loss offers the real electro-catalytically active sites for glycerol oxidation, which then maintains excellent stability, implying that Mo atoms are not the active sites for glycerol oxidation in this work. In contrast, Ratios of Ni to Mo elements in cathodic Ni-Mo-N/CFC catalyst after a short-time and a long-time electrochemical HER are the same at 1:1.29, and the corresponding concentrations of Ni and Mo elements in the electrolyte are both 0.09, i.e., almost no molybdenum was lost during the HER stability test. Thus we believe the catalysts have the robust durability for the use in this case. We have added these discussions in revised manuscript.*

7. Mechanism was proposed though the elementary and global reactions are missing. The kinetic control and thermodynamics of the related reactions should have been discussed, otherwise it is not foundational for the selectivity of the work.

Response: Thank you very much for the constructive suggestions and questions. According to the reviewer's suggestion, the experiments related to mechanism probing have been further conducted and discussed in the revised manuscript. The Faraday efficiency and selectivity of formate production have been examined at elevated potentials as shown in **Supplementary Fig. 15**. It has been found that a potential of ~ -1.35 V can be determined as the optimum potential, resulting in both high selectivity ($\sim 92.48\%$) and Faradaic efficiency ($\sim 97\%$). **Figure 3g** shows the ^1H NMR results of glycerol and product (formate) during the whole glycerol electrolysis period at 1.35 V vs RHE. The decrease of glycerol amount and the enhancement of formate concentration in the time course of electrocatalysis can be clearly observed, undoubtedly indicating the conversion of glycerol to formate. During the process of a magnitude of electric charge of ~ 385 C passing through the electrochemical cell, the concentration of formate increased to the maximum and that of glycerol decreased to 0, which suggests the complete conversion of glycerol, leading to a yield of 93% for formate production (**Figure 3h**). Surprisingly, methanol was undoubtedly detected and the ratio of methanol to formate increased to around 1:5 at ~ 129 C charges transferred, and then began to decrease until it cannot be detected. To explore the source of methanol, a three-electrode configuration using Ni-Mo-N/CFC catalyst was set up by adding 0.1 M formate dissolved in 1 M KOH at the cathode. No other products were detected except hydrogen at the cathode side (**Supplementary Fig. 18**), confirming that methanol does not come from formate reduction of cathodic reaction. Fortunately, formaldehyde was detected via phloroglucinol method, a highly sensitive method for detecting formaldehyde with a detection limit of 0.1 ppm (**Supplementary Fig. 19**). So, we infer that the methanol did come from the Cannizzaro reaction (an aldehyde without an α -hydrogen atom undergoes an intermolecular redox reaction under the action of a strong base to form a carboxylic acid and an alcohol) of formaldehyde in alkaline solution. Besides, we also found that the carbon atoms in methanol molecules came from carbon atoms at positions 1, 3 of glycerol according to the results of isotope tracer described below.

To better understand the mechanism of glycerol electro-oxidation to formate, experiments using 0.1 M 1, 3-¹³C-labeled glycerol and 0.1 M 2-¹³C-labeled glycerol in 1M KOH as electrolyte and Ni-Mo-N/CFC as electrocatalyst have been further performed at 1.35 V vs RHE (figure 4). From the ¹H NMR analysis it can be found that the ratios of unlabeled formate to ¹³C-labeled formate obtained by 2-¹³C-labeled glycerol oxidation are 0.72:1, 1.39:1, 1.9:1, and the corresponding ratios obtained by 1,3-¹³C-labeled glycerol oxidation are 1:0.82, 1:1.42, 1:1.98, respectively, during the progress of the glycerol oxidation reaction to varied stages, as determined by ¹H NMR analysis. Therefore, it is clear that the formed product, formate, comes from both the secondary and primary carbons of glycerol.

Moreover, the methanol can be detected in 1,3-¹³C-labeled glycerol, while it cannot be detected in 2-¹³C-labeled glycerol by the ¹³C NMR spectroscopy in monitoring the source of methanol (Supplementary Fig. 20), which means that the formation of formate is a consequence of successively stepped reaction after a long-time electrolysis with methanol being one of the major intermediates, as described and supposed in more detail in the following.

According to the above results and previous reports (Ref. 24, 30, 52-57), a possible reaction path of glycerol electro-oxidation at the anode to formate is proposed as shown in Scheme 1. Firstly, the formation of formate begins with glycerol oxidation to glyceraldehyde, which is then oxidized to formate and glycolaldehyde with the breakage of the C-C bonds. Next, the oxidative cleavage of the glycolaldehyde produces formate and formaldehyde, followed by the methanol and formate formations by the intermolecular redox reactions (Cannizzaro reaction) of formaldehyde in alkaline solution, and the methanol final oxidation to formate. In all, almost all of the reactant glycerol and several intermediates are eventually oxidized to formate. In this overall pathway of glycerol to formate, a very small amount of carbonate (~2.4%) may come from the further oxidation of formate.

In addition, weak peaks of glycollic acid can be detected in the ¹³C-labeled glycerol oxidation products by the ¹³C NMR spectroscopy due to the labeling of ¹³C

though they cannot be detected in the unlabeled glycerol oxidation products, indicating that the successive oxidation of glycerol to glycolic acid, and then to formate, a typical pathway of the glycerol electro-oxidation to formate, is a minor side reaction pathway in the present study.

These results and discussions have been added in the revised manuscript (Page 14-18).

8. A few glycerol concentration was tested in electrolysis, what about lower concentration? Would the lower concentration affect the stability? These experiments would benefit mechanism discussion.

Response: *Thank you very much for the constructive suggestions. According to the reviewer's suggestion, the lower concentrations of glycerol, 0.005M, 0.01M and 0.2M, have been adopted for the electrolysis tests (Supplementary Fig. 9). It has been found that the current density of glycerol oxidation decreased with the decrease of glycerol concentration. Chronopotentiometric curves of Ni-Mo-N/CFC for glycerol oxidation at these low concentrations have been also obtained (Supplementary Fig. 21). It can be seen that the potential value become gradually stabilized during the decrease glycerol concentrations from 0.1 M to 0.01M, most likely due to the oxidation balance between glycerol and the intermediate methanol. Additionally, the potential increased significantly on the chronopotentiometric curve of glycerol oxidation at 0.005 M glycerol beyond about 6 hours, indicating the later taking-place of water oxidation after the complete glycerol oxidation. These results and discussions have been added in the revised manuscript.*

9. The XPS analyses of Ni species are not convincing, in particular the Fig. 8d, why no Ni (III) and less Ni (0) were observed there? They Ni species amount can be quantified. Catalyst Vacancies were claimed to form on the electrode, though they were not proved.

Response: *Thank you very much for the constructive suggestions and questions. The fact that no Ni (III) can be observed in Fig. 4d (now in Fig. 6b) is believed to result*

from the high enough stability of Ni(II) and/or Ni(0) species in Ni-Mo-N/CFC under the reduction conditions, which made it difficult for these nickel species to be oxidized. The less observed Ni (0) in the catalyst is most probably the result of the metallic nickel conversion to Ni (II) under the presence of water vapour as demonstrated by Sargent's group, or, dissolved by oxygen in the electrolyte according to Dai's reports (Ref. 60 and 61). According to the reviewer's kind suggestion, the quantification of Ni species has been supplemented, showing the partial decrease of Ni (0) and the increase of Ni (II) amounts (Supplementary Table 4). With regarding to vacancies, thank you again for your kind suggestion. We think that the expression about the "vacancy" mentioned in the sentence "...Mo loss from the catalyst would lead to the formation of vacancy defects..." might not be accurate enough. Alternatively, it is better to state that the Mo loss from the catalyst will lead to the formation of defects, rather than vacancy, as did in a Chen's recent report. (Ref. 42) Related expressions have been corrected in the revised manuscript.

10. It stated Ni (0) on cathod was converted to Ni (II) in reaction, how could they in reducing conditions?

Response: Thank you very much for the constructive question. According to the quantification result of Ni species (Supplementary Table 4), a fraction of Ni (0) converted to Ni (II), probably due to the presence of water vapour as demonstrated by Sargent's group, or dissolved by oxygen in the electrolyte according to Dai's report (Ref. 60 and 61). These discussions have been added in the revised manuscript.

11. 90% Mo lossing was determined, why the materials still maintained morphology unchanged? why could they be stable in electrolysis? The discussion and results are not convincing, the long-term operation plausibility is dubious as well.

Response: Thank you very much for the constructive questions. Yes, 90% of molybdenum has dissolved from the anodic catalyst after the glycerol oxidation during the initial 20 cycles of CV and 3 LSV scans, however, observations show that the overall nanoplate morphology of the catalyst maintained almost unchanged. It is

noticeable that dominant Mo loss takes place in a rather short-time initial period (20 cycles of CV scans + 3LSV) of electrochemical glycerol anodic oxidation, and the ratios of Ni to Mo elements in the anodic Ni-Mo-N/CFC catalyst after a short-time and a long-time electrochemical glycerol anodic oxidation remain almost unchanged, which are 1: 0.15 and 1: 0.13, respectively, during the stability tests, and the corresponding concentrations of Ni and Mo elements in the electrolyte are 3.86 ppm and 3.17 ppm. These results indicate the excellent stability of the catalyst beyond a short period of 20 cycles of CV and 3 LSV tests during which molybdenum amount keeps little changed. Therefore it can be known that Mo atoms are not electro-catalytically active sites for glycerol oxidation in this work. In addition, only a 0.029 V potential increase, i.e., 98% of potential retention, in the glycerol oxidation was observed in 12 h CP, demonstrating very slight performance degradation during the durability test of the Ni-Mo-N/CFC catalyst (Figure 3f). These data and related discussions have been added in the revised manuscript.

12. Discussions were presented though it only repeated abstract and no critical discussions were presented, supposing the work is not completed.

Response: Thank you very much for your constructive comments. We have added abundant new experiment results, new findings and related discussions on our study, and modified the structure of the manuscript to make the logic clearer.

13. The experimental is not detailed enough and description is vague. For example the conductivity or resistance of CFC, source etc. The NH₃ treatments at 400 and 600 °C are confusing too. The ESCA and Cdl calculations should have been presented.

Response: Thank you very much for your constructive comments. According to the reviewer's suggestion, the electrochemical impedance spectra and the corresponding fitting results of CFC, NiMo-Pre/CFC and Ni-Mo-N/CFC electrodes have been obtained and discussed in the revised manuscript and supporting information (Supplementary Fig. 13 and 23). It has been found that Ni-Mo-N/CFC electrode possesses the lowest charge transfer resistance for both glycerol electro-oxidation

and HER, indicating an excellent electrical contact, extremely low impedance and fast charge transfer rate. The detailed ESCA and Cdl calculations have also been provided and discussed in the revised manuscript and supporting information (Supplementary Fig. 14 and 24). It has been found that Ni-Mo-N/CFC electrode shows a large ECSA, thus exposing abundant active sites. (Page 11, 13 and 18-20)

Reviewers' comments:

Reviewer #1 (Remarks to the Author):

In this revised manuscript, the authors have taken efforts to reply the questions I raised in the previous review. In this revision, a number of new words/sentences, and figures were added, which could improve the quality of the manuscript. This improved paper could be now accepted for publication in the Nature Communications.

Reviewer #2 (Remarks to the Author):

The authors supplied the work with additional experimental data and discussion. I find the new version of the manuscript suitable for publication without further modification.

Reviewer #3 (Remarks to the Author):

The quality of the revised version has been improved greatly, though there are still some significant issues necessitate further correction.

- 1.) It is not appropriate to claim a bifunctional catalyst for Ni-Mo-N nitride because it is not stable when working as anode. The characterizations, in particular the XPS spectra of Mo and N, disclosed that the material is corroded under glycerol oxidation. It is strange the Mo XPS of post catalyst was not presented. In all, this catalyst is not evidenced a bifunctional catalyst, so that the conclusive statements in Line 262-265 are wrong.
- 2.) The adoption of nitrides as electrocatalysts was well documented though this manuscript did not review the state of the arts for these catalysts in electrocatalysis. This aspect was raised in the first review process, although the electrocatalytic upgrading of glycerol has been justified tirelessly (which can be concise).
- 3.) The critical comments on the steam reforming is not appropriate because the major problem is not CO but other factors. The justification of formate production in glycerol oxidation should consider their concentration and cost of product purification rather than simply comparing their prices. The current arguments in the paper will mislead readers.
- 4.) The inset figures are mostly unreadable. The reaction time or duration should be clarified yet they are missed when quantitative analyses were made. For example, Line 249-251 and Figure S. 17.
- 5.) Cannizzaro reaction should be referred. The theoretical H₂ production amount was perfect matching the experimental results, which is incredible. How were the results calculated and tested?
- 6.) The discussion of TEM results conflicts with XPS and other characterizations in terms of catalyst degradation/corrosion.
- 7.) The claim of ~100% faradic efficiency is not correct as consider the corrosion of anode. This value was somehow overestimated.

Response to Reviewer 1

Comments and suggestions from Reviewer 1.

In this revised manuscript, the authors have taken efforts to reply the questions I raised in the previous review. In this revision, a number of new words/sentences, and figures were added, which could improve the quality of the manuscript. This improved paper could be now accepted for publication in the Nature Communications.

Response: We thank the reviewer for the positive comments.

Response to Reviewer 2

Comments and suggestions from Reviewer 2.

The authors supplied the work with additional experimental data and discussion. I find the new version of the manuscript suitable for publication without further modification.

Response: We thank the reviewer for the positive comments.

Response to Reviewer 3

Comments and suggestions from Reviewer 3.

The quality of the revised version has been improved greatly, though there are still some significant issues necessitate further correction.

Response: Thank you very much for the kind comments and suggestions. Please find the following detailed responses.

1. It is not appropriate to claim a bifunctional catalyst for Ni-Mo-N nitride because it is not stable when working as anode. The characterizations, in particular the XPS spectra of Mo and N, disclosed that the material is corroded under glycerol oxidation. It is strange the Mo XPS of post catalyst was not presented. In all, this catalyst is not evidenced a bifunctional catalyst, so that the conclusive statements in Line 262-265 are wrong.

Response: Thank you very much for the constructive comments and questions. Yes, strictly speaking, it is not accurate to claim that Ni-Mo-N nitride is a bifunctional catalyst because of the Mo loss when working as the anode. However, for the sake of convenience in describing the characteristics of the catalysts, electrocatalysts are customarily named as “bifunctional catalysts” when the initially identical electrocatalysts are used at both anode and cathode in spite of the possible compositional or structural changes taking place during the catalytic reactions. Such a usage can be found in a number of recent publications, for example: Angew. Chem. Int. Ed. 57, 15237-15242 (2018); J. Mater. Chem. A 5, 13648-13658 (2017); J. Mater. Chem. A 6, 8479-8487, (2018), et.al. Therefore, to be in consistence with the general customary usage of such an expression, we still prefer to use the word “bifunctional” in the revised manuscript but with a quotation mark, indicating the initially identical catalyst but with compositional changes during operation, as have been well discussed in the text. In addition, the Mo XPS spectrum of the used catalyst has been presented in the revised supplementary information (Supplementary Fig. 27) and discussed in the revised manuscript (page 22). At lines 262-265 of the last-revised

manuscript version we described the stability of Ni-Mo-N/CFC. Although a large amount of Mo has been indeed corroded at the initial period of glycerol anodic electrocatalysts, afterwards the corroded Ni-Mo-N/CFC shows stable performance towards glycerol oxidation as confirmed by the U-t curves (Figure. 3f).

2. The adoption of nitrides as electrocatalysts was well documented though this manuscript did not review the state of the arts for these catalysts in electrocatalysis. This aspect was raised in the first review process, although the electrocatalytic upgrading of glycerol has been justified tirelessly (which can be concise).

Response: *Thank you for the constructive suggestion. Indeed, most transition metal-based nitrides, which usually have relatively low electrical resistance and mechanical stability, are considered as promising candidates of electrocatalysts and show efficient activity in a variety of reactions such as OER (Ref. 32), HER (Ref. 33) and ORR (Ref. 34). However, as far as we can know, there is still no report on nitrides as “bifunctional” electrocatalyst for glycerol electrooxidation and HER so far. This information has been added in the revised manuscript (page 5). The introduction about electrocatalytic upgrading of glycerol has been simplified in the revised manuscript.*

3. The critical comments on the steam reforming is not appropriate because the major problem is not CO but other factors. The justification of formate production in glycerol oxidation should consider their concentration and cost of product purification rather than simply comparing their prices. The current arguments in the paper will mislead readers.

Response: *Thank you very much for the constructive comment and suggestion. Yes, we also agree that CO emission is not the major problems in steam reforming. Methane is a non-renewable source, and the methane steam reforming, one of the most widely used process for hydrogen production, is an unsustainable strategy due to the massive energy consumptions and the simultaneous CO₍₂₎ emission as a byproduct during the thermal transformation of fossil fuels and the reforming, which*

is severely unfavorable for its applications. We have corrected related expressions in the revised manuscript (page 3).

Thank you for your construction suggestion about the justification of formate production in glycerol oxidation. The statement on the justification of formate production in glycerol oxidation has been re-edited in the revised manuscript (page 4). Formate (or formic acid) is an important chemical intermediate in large numbers of industrial processes, which has attracted great interest in the field of fuel cell applications and can be used for hydrogen storage due to its relatively high hydrogen capacity. Traditional industrial method for producing formic acid, methyl formate hydrolysis, is a complicated multi-step process that consumes a large amount of energy, and the raw materials (toxic CO) used for this process usually require gasification of coal and natural gas at elevated temperatures, which makes the high cost and consequent high price of formic acid (approximately \$1300/ton), considerably higher than those of glycerol. Considering that glycerol is a low value by-product during the production of biodiesel, electrochemical glycerol oxidation is apparently highly potential for formic acid production.

Naturally, the production's concentration and cost of purification should be taken into account in actual production applications. In the purification process, the obtained formate-electrolyte (stream 1) can be separated into pure product (stream 2) and waste (stream 3) via liquid-liquid extraction [Nat. Energy 4, 466-474 (2019)]. The minimum work of separation value (W_{min} / KJ) which is closely related to the cost can be calculated using the following equation (1):

$$W_{min} = -RT \left(N_1 \sum_{k=1..n} X_{1,k} \ln X_{1,k} - N_2 \sum_{k=1..n} X_{2,k} \ln X_{2,k} - N_3 \sum_{k=1..n} X_{3,k} \ln X_{3,k} \right) \quad (1)$$

Where R is the universal gas constant ($8.314 \text{ J mol}^{-1} \text{ K}^{-1}$), T is the temperature (298 K), N_j (where $j = 1, 2$ or 3) denotes the molar flow rate of stream j , and $X_{j,k}$ denotes the molar concentration of substance k in stream j . The W_{min} is very sensitive to the

concentration of formate ($X_{I, product}$) (a near-exponential relationship) according to the above formula. Fortunately, the formate-electrolyte can be concentrated via evaporating away water using cheaper energy input such as solar energy to increase $X_{I, product}$, thereby reducing the cost of purification. Therefore, although the exact cost cannot be given, considering concentrated formate electrolyte at a low cost and the large price gap between glycerol and formate, significant potentials of electrochemical glycerol oxidation to produce formate can be expected even the concentration and purification of produced formate is taken into account. And we will pay attention to the concentrations and purification processes in the following researchs. To avoid misleading readers, these information have been added in the revised manuscript.

4. The inset figures are mostly unreadable. The reaction time or duration should be clarified yet they are missed when quantitative analyses were made. For example, Line 249-251 and Figure S. 17.

Response: Thank you very much for the constructive suggestions. According to the reviewer's suggestion, all the inset figures (**Figure 2b-c, Figure 3f, Figure 4, Figure 7b, Figure S. 1, Figure S. 13, Figure S. 20 and Figure S. 23**) have been re-edited in the revised manuscript and supporting information.

The quantitative determination of carbonate and formate (**Line 249-251 and Figure S. 17**) was carried out after glycerol oxidation for 12h. These information has been added in the revised manuscript and supporting information (**page 14 and 30**). Additionally, we have supplemented reaction time periods where it is necessary in the revised manuscript.

5. Cannizzaro reaction should be referred. The theoretical H₂ production amount was perfect matching the experimental results, which is incredible. How were the results calculated and tested?

Response: Thank you very much for the constructive suggestions and questions. According to the reviewer's suggestion, references about the cannizzaro

reaction has been cited in the revised manuscript (*J. Am. Chem. Soc.* **101**, 3576-3583 (1979); *Justus Liebigs Ann. Chem.* 88, 129 (1853), **Ref. 53 and 54**).

H_2 produced at the cathode was measured by a gas chromatograph (Ramiin GC2060) equipped with a packed column and a thermal conductivity detector (TCD), quantified by the external standard method. The standard curve of H_2 has been obtained and exhibited in **Supplementary Fig.31**. For each measurement, samples of 0.3 mL were collected from the same position of sealed cell and injected into the GC instrument carefully to determine the amount of H_2 produced. The theoretically generated H_2 amount was calculated following $Q_{tot} \times V_m / (Z \times F)$ (Q_{tot} is the total charge passed through the electrodes, V_m is the molar volume of gas, $Z = 2$ is the number of electrons needed to produce a molecule of H_2 , $F = 96485 \text{ C mol}^{-1}$ is the Faraday constant). As shown in **Figure 7d**, when 12.09 C charge calculated by electrochemical workstation passed through the electrochemical cell, 1.506 mL H_2 measured by GC was generated on the cathode, which is in close accordance with the theoretical value of 1.516 mL. These information has been added in the revised manuscript (**page 31**).

6. The discussion of TEM results conflicts with XPS and other characterizations in terms of catalyst degradation/corrosion.

Response: Thank you for your constructive comment. Yes, the description of TEM is not accurate enough in the text and we have corrected it as shown in **page (22)** in revised manuscript.

The discussion “overall oxidations of Ni (0) and/or Ni (II) to higher valence state Ni species” in manuscript is not appropriate due to the small amount of remains of metallic Ni as shown in HRTEM image (Figure 6e), as proved in the results of XRD (Supplementary Fig. 26) and XPS (Figure 6a). To be more accurate, the revised expression of “oxidations of most Ni (0) and/or Ni (II), rather than overall oxidations, to higher valence state Ni species”, has been used (**page 21**) in the revised manuscript.

Thank you for your construction suggestion again and we will examine the

manuscript more closely to make the discussion and results more accurate.

7. The claim of ~100% faradic efficiency is not correct as consider the corrosion of anode. This value was somehow overestimated.

Response: *Thank you for your constructive comments. In this work, the Faraday efficiencies (FE) of close to 100% for H₂ evolution at cathode and 95% for formate production at anode were obtained, respectively. The corrosion of anode should be considered in FEs calculation. During the glycerol oxidation at the anode, the charge consumed by molybdenum dissolution is 1.52 C, which accounts for only 0.4 % of the total 385 C charge required for the complete oxidation of glycerol to formate. Therefore, the effect of anodic corrosion on the FE of formate production is insignificant, and we believe that the 95% FE of formate production should be rational that in this work.*

REVIEWERS' COMMENTS:

Reviewer #3 (Remarks to the Author):

The questions raised by the reviewer were clearly answered and most of them are acceptable. I would like to leave it to editor to decide the acceptance because I am not available to review further because of some urgent tasks.

I personally would like to recommend it being accepted for publication, but strongly recommend the authors and editor to think over:

1.) the "Bifunctional" terms are removed from Title and corrected in the full text. A publication is to deliver reasonable science rather than "Citations". In the scientific community, there are already too many misleading fancy words applied. At least this "bifunctional" phrase misleads readers and disseminates "misappropriate" science.

2.) I am not confident with the calculated 100% and 95% Faradic efficiency numbers for individual electrodes. corrosion contributed electrons to cathodic reaction, which leads 100% untrustable. Further, do these efficiency numbers follow the charge conservation law in their system?

I declare here I don't have any interest conflicts relevant to this paper.

Response to Reviewer 3

Comments and suggestions from Reviewer 3.

The questions raised by the reviewer were clearly answered and most of them are acceptable. I would like to leave it to editor to decide the acceptance because I am not available to review further because of some urgent tasks.

I personally would like to recommend it being accepted for publication, but strongly recommend the authors and editor to think over:

Response: Thank you very much for the recommendation. We have considered the kind suggestions and questions carefully and please find the following detailed responses.

1. The "Bifunctional" terms are removed from Title and corrected in the full text. A publication is to deliver reasonable science rather than "Citations". In the scientific community, there are already too many misleading fancy words applied. At least this "bifunctional" phrase misleads readers and disseminates "misappropriate" science.

Response: Thank you very much for the constructive comments and suggestions. We are now aware that the term "bifunctional" is not appropriate as Ni-Mo-N nitride catalyst at the anode has experienced significant composition change during the electrocatalysis. According the reviewer's kind suggestion, we have removed the term "bifunctional" from the title and corrected in the full text.

2. I am not confident with the calculated 100% and 95% Faradic efficiency numbers for individual electrodes. Corrosion contributed electrons to cathodic reaction, which leads 100% untrustable. Further, do these efficiency numbers follow the charge conservation law in their system?

Response: Thank you very much for the comment. Yes, our efficiency numbers follow the charge conservation law. According to the ICP-OES analysis (Supplementary Table 1 and 2), a limited amount of Mo species was dissolved from the Ni-Mo-N/CFC

electrode at the cathode during the initial short time period (20 cycles of CV scans+ 3LSV) of electrochemical hydrogen evolution reaction, after which the no Mo loss took place from the catalyst. In order to ensure the accuracy of data as much as possible, a Faraday efficiency (FE) value of 99.7 %, which is close to 100%, for H₂ evolution was measured after 20 cycles of CV scans and 3 LSV tests, when the catalyst kept stable without further Mo dissolution.

As for the anode, the maximum charge consumed by molybdenum dissolution would be as low as 1.52 C even if all molybdenum species was dissolved during glycerol anodic oxidation as indicated in the last revision, which accounts for only 0.4 % of the total 385 C charge required for the complete oxidation of glycerol to formate. The actual calculation indicates that a FE value of 95.0% for formate production was obtained after 20 cycles of cyclic voltammetry scans and 3 LSV of electrochemical glycerol anodic oxidation. According to the ICP-OES analysis, molybdenum dissolution process almost ended in the 20 cycles of cyclical voltammetry and 3 LSV tests, after that the amount of Mo in the catalyst and electrolyte remained almost unchanged (Supplementary Table 1 and 2). This fact means that the obtained FE value of 95.0% in our experiments is the one almost after the completion of Mo dissolution.

Therefore, we believe that the 95.0% and 99.7% FE values respectively for formate and H₂ productions are reasonable in this work. Related discussions have been added in the revised manuscript.